# NVFi: Neural Velocity Fields for 3D Physics Learning from Dynamic Videos

**Jinxi Li    Ziyang Song    Bo Yang**

vLAR Group, The Hong Kong Polytechnic University

jinxi.li@connect.polyu.hk    ziyang.song@connect.polyu.hk    bo.yang@polyu.edu.hk

## Abstract

In this paper, we aim to model 3D scene dynamics from multi-view videos. Unlike the majority of existing works which usually focus on the common task of novel view synthesis within the training time period, we propose to simultaneously learn the geometry, appearance, and physical velocity of 3D scenes only from video frames, such that multiple desirable applications can be supported, including future frame extrapolation, unsupervised 3D semantic scene decomposition, and dynamic motion transfer. Our method consists of three major components, 1) the keyframe dynamic radiance field, 2) the interframe velocity field, and 3) a joint keyframe and interframe optimization module which is the core of our framework to effectively train both networks. To validate our method, we further introduce two dynamic 3D datasets: 1) Dynamic Object dataset, and 2) Dynamic Indoor Scene dataset. We conduct extensive experiments on multiple datasets, demonstrating the superior performance of our method over all baselines, particularly in the critical tasks of future frame extrapolation and unsupervised 3D semantic scene decomposition. Our code and data are available at https://github.com/vLAR-group/NVFi

## 1 Introduction

The 3D world around us is constantly changing over time, where objects are falling, vehicles moving, and clocks ticking. Humans can effortlessly learn the geometry and physical properties of such dynamic 3D scenes, and further predict their future motions following the learned physics rules, just by watching the things for a few seconds. Giving machines such ability to automatically infer the geometry and physics of complex dynamic 3D scenes is essential for many cutting-edge applications in augmented reality, games, and the movie industry. Recent advances in the emerging area of neural radiance field [37] and its succeeding variants [46, 4, 6, 19, 75, 40] have shown excellent results in modeling dynamic 3D scenes such as deformable things [4, 58] and articulated objects [39, 55]. While showing superior performance in interpolating novel views within the observed time period, almost all these methods tend to fit training image sequences, without explicitly learning the physical properties such as object velocities, thus being unable to extrapolate and predict future motion patterns of 3D scenes.

More recently, a few studies [47, 14, 8, 32, 2] start to integrate physics priors into implicit neural representations to model dynamic 3D scenes such as floating smoke or simple moving objects. By introducing the governing PDEs, *a.k.a.*, PINN [48], these methods demonstrate promising reconstruction of 3D scene geometry, appearance, velocity and/or viscosity fields. However, the learned physical properties are either tightly coupled with the target objects [14] in the scene or require additional foreground segmentation masks in the loop [32]. This means that the estimated physics knowledge from dynamic video frames is not clearly disentangled, thereby not transferable from one scene to another.

37th Conference on Neural Information Processing Systems (NeurIPS 2023).

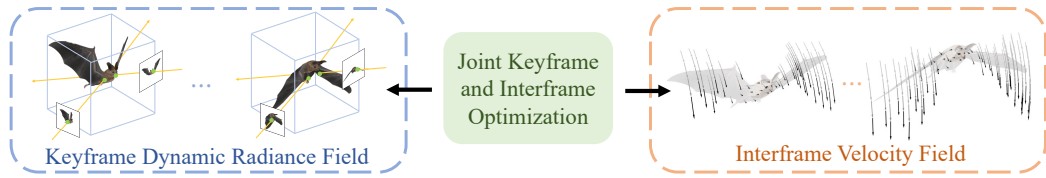

Figure 1: The three major components of our framework: Keyframe Dynamic Radiance Field, Interframe Velocity Field, and the Joint Keyframe and Interframe Optimization module.

In this regard, we ask an ambitious question: *Can we learn disentangled physical properties alongside recovering geometry and appearance of dynamic 3D scenes purely from multi-view video frames?* Among various physical properties in the scene, we choose to focus on velocity as it primarily governs scene movement dynamics. Such disentangled velocity fields, once successfully learned, are expected to unlock multiple desirable applications: 1) Future frame extrapolation and prediction beyond the observed time period in training. For example, after watching a football flying in the penalty area, we can predict what will happen next. 2) Dynamics transformation from one scene to another. For instance, after watching a bird flipping wings, we can imagine the same physical behavior on the body of an airplane. 3) Semantic decomposition of 3D scenes. Intuitively, once the velocity field of an entire dynamic 3D scene is learned, all individual objects or parts undergoing different moving patterns can be easily segmented, without needing any extra human annotations for training.

However, accurately learning the physical velocity of a whole 3D scene space is particularly challenging, primarily due to the lack of ground truth 3D velocity annotations, the unknown object types or materials in the scene, and the sparse yet long-time visual trajectory in training. In addition, when separately learning the velocity field, it is usually an under-constrained problem even with the commonly used PINN technique [14].

To tackle these challenges, we introduce a new general framework to simultaneously model the geometry, appearance, and disentangled velocity of a dynamic 3D scene only from multi-view video frames. In particular, as shown in Figure 1, our framework consists of three major components: 1) a **keyframe dynamic radiance field** to learn time-dependent volume density and appearance for every 3D point in space; 2) an **interframe velocity field** to learn time-dependent 3D velocity for every point as well; and 3) a **joint keyframe and interframe optimization** method together with physics informed constraints to train both networks. For the first component, it is flexible to adopt any of the existing time-dependent NeRF architectures such as HexPlane [6] or K-Planes [19]. For the second component, the neural network actually can be as simple as MLPs.

The core of our framework is the third component, where we explicitly apply three types of loss functions to jointly optimize both networks: 1) the keyframe photometric loss, 2) the interframe photometric loss, and 3) the governing PDE losses. With these losses, our framework can precisely learn disentangled velocity fields, without needing additional regularization terms on volume density or information on object masks, types, or materials. Overall, our framework models general dynamic 3D scenes by learning **n**eural **v**elocity **fi**elds with physical priors. Our method is named **NVFi** and our contributions are:

- We introduce a general framework to model dynamic 3D scenes as physics-informed radiance fields from multi-view videos, without requiring information on object types, materials, or masks.
- We design a neural velocity field together with a joint keyframe and interframe optimization method to effectively train the networks.
- We demonstrate three applications for the learned velocity fields on two newly collected dynamic 3D datasets and a challenging real-world dataset, showing superior results in future frame extrapolation, semantic decomposition, and velocity transferring across 3D scenes.

## 2   Related Works

**Static 3D Representations:** Conventional representations for static 3D objects and scenes include voxels [13, 71, 72], point clouds [17], octrees [57, 64], meshes [28, 23], and primitives [77]. Due to the discretization issue of these explicit representations, the fidelity of 3D shapes is usually limited by spatial resolution and memory footprint. Inspired by the seminal works [10, 36, 41], recently, there

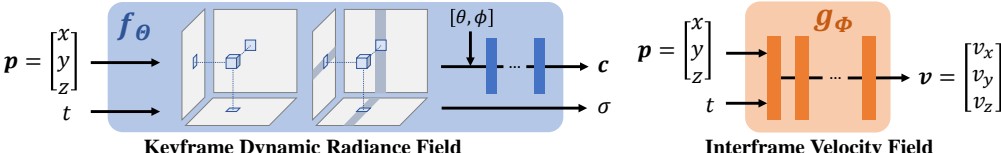

Figure 2: The left block illustrates the network architecture of our keyframe dynamic radiance field based on HexPlane [6], and the right block shows the architecture of our interframe velocity field.

has been a strong interest in representing 3D data in implicit neural functions. These methods are generally classified as: 1) occupancy fields (OF) [10, 36], 2) signed distance fields (SDF) [41], 3) unsigned distance fields (UDF) [12, 62], and 4) radiance fields (NeRF) [37, 3, 59]. Basically, these neural representations simply take 3D point locations as input and directly regress the corresponding point information, such as occupancy, distance to surface, color, semantics [76, 61], *et al.*, via MLPs. Compared with traditional explicit representations, these implicit neural representations allow for continuous shape and appearance modeling at a particularly low memory footprint.

**Dynamic 3D Representations:** Recent techniques in this category are primarily built on the appealing NeRF architecture [37], thanks to its unprecedented level of fidelity in representing various 3D objects and scenes [6, 19]. Given video sequences of dynamic scenes, these NeRF based methods usually take the time $t$ as an additional input dimension and then optimize the entire network using the standard photometric loss. Generally, the techniques can be classified as: 1) deformation-based methods [42, 58, 4, 46, 5, 18, 74], 2) flow-based methods [21, 33, 16, 63, 35], and 3) direct space-time based methods [69, 1, 31, 43]. In parallel, there are also a number of domain-specific NeRFs to model particular dynamic objects such as human bodies [44, 50, 45, 70, 68] and faces [4, 20, 66, 67]. Although all these methods have shown excellent performance in novel view synthesis on a wide range of dynamic datasets, they are primarily designed to interpolate frames within the time period of training data, lacking the ability to predict future frames. By contrast, our NVFi aims to model the underlying physical principles of 3D scenes, thus being easily able to extrapolate future dynamics.

**Physics Informed Deep Learning:** Unlike traditional numerical methods such as finite element methods, recent physics-informed neural networks (PINN) [48] represent tensor fields with neural networks, converting the PDE solutions into optimizing network weights via PDE-based loss functions. Inspired by the pioneering works [48, 52, 30], a plethora of succeeding works [51, 49, 25, 38, 65], have been developed, demonstrating excellent results in a wide range of applications such as acoustics [53, 56], fluids [60, 22, 15], 3D scene representations [8, 14, 32, 47], *et al.*, thanks to its mesh-free formulation. More details can be found in the recent surveys [27, 24]. In our framework, we find that the commonly-used PINN constraints are insufficient to learn accurate 3D velocity fields. To tackle this issue, we further propose the keyframe based velocity propagation module.

## 3 NVFi

### 3.1 Overview

As shown in Figure 2, our framework consists of two neural networks together with their optimization methods. Given a set of images of a dynamic scene with known camera poses and intrinsics, the **keyframe dynamic radiance field** $f_\Theta$ simply takes a 3D point $p = (x, y, z)$, viewing angle $(\theta, \phi)$, and timestamp $t$ as input, directly regressing the volume density $\sigma$ and color $c = (r, g, b)$. For the network architecture, we simply adopt the recent HexPlane [6] which shows excellent performance in efficiently modeling dynamic video frames, though other NeRF variants can also be used. Formally, our keyframe dynamic radiance field is defined below and implementation details are in Appendix.

$$(\sigma, c) = f_\Theta(x, y, z, \theta, \phi, t) \quad \Theta \text{ are trainable parameters.} \tag{1}$$

For the **interframe velocity field** $g_\Phi$, it takes the 4D vector $(x, y, z, t)$ as input, and predicts the 3D velocity $v = (v_x, v_y, v_z)$ for point $p$ at time $t$. For simplicity, the network $g_\Phi$ is parameterized by simple MLPs, though more advanced architectures can be applied as well. Formally, the velocity field is defined below and implementation details are in Appendix.

$$v = g_\Phi(p, t) = g_\Phi(x, y, z, t) \quad \Phi \text{ are trainable parameters of MLPs.} \tag{2}$$

With these two networks and training images sampled from a dynamic 3D scene, the key challenge is to effectively optimize these networks, such that the final learned velocity field is precise and disentangled, supporting future frame extrapolation, motion transfer, and semantic decomposition.

## 3.2 Optimization of Keyframe Dynamic Radiance Fields

Given dynamic video frames of $T$ timestamps $\{1 \cdots t \cdots T\}$, there are two potential strategies to optimize dynamic neural radiance fields.

- **Strategy 1 - Dense Frame Optimization:** This strategy uses all available video frames of a specific dynamic 3D scene in the training set to optimize a dynamic radiance field. This means that for the network $f_\Theta$, the time dimension $t$ is densely sampled during optimization. Formally:

$$f_\Theta(x, y, z, \theta, \phi, t): \quad t \in \{1 \cdots t \cdots T\} \tag{3}$$

However, it has two limitations: 1) It is inefficient to learn accurate 3D geometry and appearance because this strategy is somewhat equivalent to modeling a dense number of static radiance fields for all timestamps. 2) It is hard to obtain a disentangled physical velocity field for the entire 3D scene, as the change of physical geometries is tightly encoded within the dynamic radiance field.

- **Strategy 2 - Canonical Frame Optimization:** This strategy optimizes a canonical representation of 3D geometry and appearance, usually joined with another deformation or transportation network to warp the future $(T-1)$ timestamps back to the first timestamp. It can be seen as:

$$f_\Theta(x, y, z, \theta, \phi, t): \quad t \in \{1\} \tag{4}$$

However, such a strategy has a strong assumption that the corresponding point appearances across different timestamps keep unchanged. Therefore, for dramatically changing 3D scenes, the optimized geometry, appearance, and possible jointly learned physics properties tend to be inferior.

**Our Strategy - Keyframe Optimization:** In this regard, we propose to adopt a keyframe based strategy to learn the dynamic radiance field $f_\Theta$. In particular, we uniformly sample $K$ timestamps out of the total $T$ stamps to optimize $f_\Theta$. Formally:

$$f_\Theta(x, y, z, \theta, \phi, t_k): \quad t_k \in \left\{ [T/K], 2[T/K], 3[T/K], \cdots T \right\} \tag{5}$$

Color images are rendered from the above keyframe dynamic radiance fields by sampling points along rays. For each pixel, *i.e.* ray $r$, in a keyframe at time $t_k$, the appearance $C(r, t_k)$ is obtained by volume rendering of NeRF [37]. Then the network $f_\Theta$ can be optimized by the following keyframe photometric loss. Details of the rendering equation are in Appendix.

$$\ell_{keyframe} = ||C(r, t_k) - \bar{C}(r, t_k)||, \quad \text{where } \bar{C}(r, t_k) \text{ is the ground truth color observation.} \tag{6}$$

This keyframe optimization strategy, albeit simple, has two unique advantages: 1) It allows to sufficiently and accurately fit the sparsely sampled dynamic 3D scene geometry and appearance given the same network capacity. 2) It allows the remaining interframes belonging to the $(T-K)$ time stamps to be used for optimizing a disentangled velocity field, as discussed in Section 3.3.

## 3.3 Optimization of Interframe Velocity Fields

As to our separate interframe velocity field $g_\Phi$, it is impossible to directly supervise it using ground truth labels as they cannot be collected in practice. However, there are some physics rules to regularize the velocity field. The objects are transported by the velocity field, and the velocity field is transported by itself according to some unobservable hidden forces. In order to keep the mass and appearance of the objects, the velocity field needs to be divergence-free and obeys the basic law of momentum conservation, whose details are in Appendix. Note that, more complex physics dynamics beyond the daily 3D scenarios are out of the scope of this paper. In this regard, our velocity field, *i.e.*, $v = g_\Phi(p, t)$, needs to firstly satisfy the following two constraints.

$$\nabla_p \cdot v = 0, \qquad \frac{\partial v}{\partial t} + v \cdot \nabla_p v = a \tag{7}$$

We simply turn these PDEs into the following two PINN losses [48] to optimize the velocity field. Here we use $\boldsymbol{v}(\boldsymbol{p},t)$ as the network to avoid abuse of notation.

$$\ell_{divergence\_free} = \frac{1}{NM} \sum_{n=1}^{N} \sum_{m=1}^{M} ||\nabla_{\boldsymbol{p}_n} \cdot \boldsymbol{v}(\boldsymbol{p}_n, t_m)||$$

$$\ell_{momentum} = \frac{1}{NM} \sum_{n=1}^{N} \sum_{m=1}^{M} ||\frac{\partial \boldsymbol{v}(\boldsymbol{p}_n, t_m)}{\partial t_m} + \boldsymbol{v}(\boldsymbol{p}_n, t_m) \cdot \nabla_{\boldsymbol{p}_n} \boldsymbol{v}(\boldsymbol{p}_n, t_m) - \boldsymbol{a}|| \quad (8)$$

where $\boldsymbol{p}_n$ is uniformly sampled in the whole 3D scene volume, and $t_m$ is uniformly sampled from 0 to the interested maximum extrapolation time $t_{max}$, and $\boldsymbol{a}$ is the general acceleration term learned by another MLP-based network: $(x,y,z,t) \rightarrow \boldsymbol{a}$, whose details are in Appendix. Nevertheless, with such PINN losses, it is insufficient to optimize the velocity field itself, since there are infinitely many solutions. To tackle this, we introduce an additional interframe optimization strategy.

**Interframe Optimization Strategy:** Naturally, the 3D scene geometry and appearance encoded in the keyframe dynamic radiance field, once appropriately transported by the velocity field, should be able to render 2D images to match with the ground truth observations in interframes belonging to the remaining $(T - K)$ timestamps. To enforce such a constraint, the key challenge is to determine the color and density values for all 3D points at each interframe timestamp, such that the volume rendering equation can be applied to estimate RGB for each pixel at the interframe timestamp, after which the photometric loss can be adopted. To tackle this, we propose the following Algorithm 1.

---

**Algorithm 1** At a specific interframe timestamp $t_i$, given a light ray $\boldsymbol{r}_i$ with viewing angle $(\theta, \phi)$ and $S$ sample points $\{\boldsymbol{p}_1 \cdots \boldsymbol{p}_s \cdots \boldsymbol{p}_S\}$ along the ray, the objective of this algorithm is to determine the color and density values for the $S$ points along $\boldsymbol{r}_i$, denoted as: $\{(\boldsymbol{c}_1, \sigma_1) \cdots (\boldsymbol{c}_s, \sigma_s) \cdots (\boldsymbol{c}_S, \sigma_S)\}$. In the meantime, we also have the keyframe dynamic radiance field $f_\Theta$ and velocity field $g_\Phi$.

Note that, we shall not directly query $f_\Theta$ to obtain color and density for the $S$ points because: 1) the dynamic radiance field $f_\Theta$ is never trained on interframe timestamps, thus the queried values are inaccurate; 2) the velocity field $g_\Phi$ will not be involved, and therefore the interframes cannot provide additional constraints to optimize $g_\Phi$.

**Input:**
- The ray direction $(\theta, \phi)$, the interframe timestamp $t_i$, the $S$ sample points on the ray $\{\boldsymbol{p}_1 \cdots \boldsymbol{p}_s \cdots \boldsymbol{p}_S\}$;
- The $K$ keyframe timestamps $\{t_1 \cdots t_k \cdots t_K\}$;
- The initialized and ongoing training networks: $f_\Theta$ and $g_\Phi$;

**Output:**
- The color and density values for $S$ sample points along the ray: $\{(\boldsymbol{c}_1, \sigma_1) \cdots (\boldsymbol{c}_s, \sigma_s) \cdots (\boldsymbol{c}_S, \sigma_S)\}$;

**Preliminary step:**
- Find the nearest keyframe timestamp $\hat{t}_k$ for the interframe timestamp $t_i$:

$$\hat{t}_k = \arg\min_{t_k} |t_k - t_i|$$

**for** $\boldsymbol{p}_s$ in $\{\boldsymbol{p}_1 \cdots \boldsymbol{p}_s \cdots \boldsymbol{p}_S\}$ **do**
- Transport $\boldsymbol{p}_s$ to its corresponding point $\boldsymbol{p}'_s$ at its nearest keyframe timestamp $\hat{t}_k$, according to its velocity field. The position of $\boldsymbol{p}'_s$ can be obtained by:

$$\boldsymbol{p}'_s = \boldsymbol{p}_s + \int_{t_i}^{\hat{t}_k} g_\Phi(\boldsymbol{p}_s(t), t) dt, \quad \text{Runge-Kutta 2 solver [9] is applied in our implementation.}$$

- Retrieve the volume density $\sigma'_s$ and view-agnostic color feature vector $\boldsymbol{e}'_s$ for point $\boldsymbol{p}'_s$:

$$(\sigma'_s, \boldsymbol{e}'_s) \leftarrow f_\Theta(\boldsymbol{p}'_s, \hat{t}_k), \quad \text{Note: HexPlane backbone [6] can output a view-agnostic color feature vector } \boldsymbol{e}'_s.$$

- Assign the retrieved features of $\boldsymbol{p}'_s$ to the original point $\boldsymbol{p}_s$: $(\sigma_s, \boldsymbol{e}_s) \leftarrow (\sigma'_s, \boldsymbol{e}'_s)$.
- Obtain the color $\boldsymbol{c}_s$ for point $\boldsymbol{p}_s$:

$$\boldsymbol{c}_s \leftarrow \tilde{f}_\Theta(\boldsymbol{e}_s, \theta, \phi), \quad \text{Note: } \tilde{f}_\Theta \text{ is a subnetwork of HexPlane backbone [6] as detailed in Appendix.}$$

- Output $(\boldsymbol{c}_s, \sigma_s)$ for point $\boldsymbol{p}_s$.

After the above *for loop*, we obtain all color and density values for $S$ sample points.

---

Having the color and density values of all 3D points along ray $\boldsymbol{r}_i$ in the interframe of timestamp $t_i$, the appearance $\boldsymbol{C}(\boldsymbol{r}_i, t_i)$ is obtained by volume rendering of NeRF [37]. Then both networks $f_\Theta$ and $g_\Phi$ can be optimized by the following interframe photometric loss.

$$\ell_{interframe} = ||\boldsymbol{C}(\boldsymbol{r}_i, t_i) - \bar{\boldsymbol{C}}(\boldsymbol{r}_i, t_i)||, \quad \text{where } \bar{\boldsymbol{C}}(\boldsymbol{r}_i, t_i) \text{ is the ground truth color observation.} \quad (9)$$

### 3.4 Joint Keyframe and Interframe Optimization

Technically, the keyframe dynamic radiance field $f_\Theta$ can be firstly trained by $\ell_{keyframe}$ only, and then the interframe velocity field $g_\Phi$ by $\ell_{divergence\_free} + \ell_{momentum} + \ell_{interframe}$.

Nevertheless, we empirically find that simultaneously propagating errors of interframes back to the radiance field $f_\Theta$ helps achieve better performance overall. Therefore, we adopt the following joint keyframe and interframe strategy to optimize both networks.

$$f_\Theta \leftarrow (\ell_{keyframe} + \ell_{interframe}) \qquad g_\Phi \leftarrow (\ell_{divergence\_free} + \ell_{momentum} + \ell_{interframe}) \qquad (10)$$

## 4 Experiments

**Datasets:** Our framework primarily focuses on learning meaningful physical velocity fields for dynamic 3D scenes, instead of simply fitting video frames. Although there are a number of dynamic 3D scene datasets in the literature, they are mainly collected for the popular task of novel view synthesis within the training time period, *i.e.*, interpolation in time dimension. Besides, the underlying motions of these scenes tend to be chaotic, and estimating their future motions or transferring their motions are hardly meaningful or entertaining in practice. In this regard, we introduce two new synthetic datasets: 1) Dynamic Object dataset, and 2) Dynamic Indoor Scene dataset.

*1) Dynamic Object Dataset:* This dataset consists of 6 distinct 3D objects, each of which displays a unique motion pattern, including either rigid or deformable movements in 3D space. These 3D objects and their realistic motions are all designed by unknown external practitioners from SketchFab, and we purchased their Licenses and will make them available for free use in the community.

For each 3D object, we collect RGB images at 15 different viewing angles over 1 second, where each viewing angle has 60 frames captured. We reserve the first 45 frames at randomly picked 12 viewing angles as the training split, *i.e.*, 540 frames, while leaving the 45 frames at the remaining 3 viewing angles for testing interpolation ability, *i.e.*, 135 frames for novel view synthesis within the training time period, and keeping the last 15 frames at all 15 viewing angles for evaluating future frame extrapolation, *i.e.*, 225 frames. More details are in Appendix.

*2) Dynamic Indoor Scene Dataset:* We also collect another synthetic dynamic 3D dataset, which includes 4 indoor scenes with multiple complex 3D objects undergoing different rigid body motions. There are about 4 objects such as tables or chairs in each 3D scene. Basically, such an indoor dataset aims to simulate potential scenarios for robotics or VR applications to understand dynamic 3D scenes.

Since the indoor scene is significantly more challenging, for each 3D scene, we collect RGB images at 30 viewing angles over 1 second, where each viewing angle has 60 frames captured. Similarly, we reserve the first 45 frames at randomly picked 25 viewing angles as the training split, *i.e.*, 1125 frames, while leaving the 45 frames at the remaining 5 viewing angles for testing interpolation ability, *i.e.*, 225 frames, and keeping the last 15 frames at all 30 viewing angles for evaluating future frame extrapolation, *i.e.*, 450 frames. The ground truth object segmentation masks are also collected for evaluating the semantic decomposition capability in Section 4.2. More details are in Appendix.

While existing dynamic 3D scene modeling techniques and the commonly used datasets in literature are mainly designed for novel view rendering/interpolation within the training time period, rather than for extrapolation beyond the training time period, we evaluate our method on two selected scenes from a real-world dataset: NVIDIA Dynamic Scene[73]. It captures real-world dynamic scenes by a static camera rig with 12 cameras. For each scene, we clip 60 frames with reasonable and predictable motions. We reserve the first 46 frames at randomly picked 11 cameras as the training split, *i.e.*, 506 frames, while leaving the 46 frames at the remaining 1 camera for testing interpolation ability, *i.e.*, 46 frames for novel view synthesis within the training time period, and keeping the last 14 frames at all 12 cameras for evaluating future frame extrapolation, *i.e.*, 168 frames.

**Baselines:** We carefully choose three representative groups of methods as our baselines: 1) dense frame optimization method T-NeRF [46] and NSFF [33], 2) canonical frame optimization methods D-NeRF [46] and TiNeuVox [18], 3) PINN methods T-NeRF$_{PINN}$ and HexPlane$_{PINN}$. Both methods are adapted by us via integrating a separate velocity field supervised by the same PINN losses as ours.

**Metrics:** For evaluating both interpolation and future frame extrapolation and motion transfer, the standard metrics **PSNR**, **SSIM**, and **LPIPS** scores are reported across testing views. For evaluating

Table 1: Quantitative results of all methods for both novel view interpolation and future frame extrapolation on Dynamic Object Dataset and Dynamic Indoor Scene Dataset.

| | Dynamic Object Dataset | | | | | | Dynamic Indoor Scene Dataset | | | | | |
| --- | --- | --- | --- | --- | --- | --- | --- | --- | --- | --- | --- | --- |
| | Interpolation | | | Extrapolation | | | Interpolation | | | Extrapolation | | |
| | PSNR↑ | SSIM↑ | LPIPS↓ | PSNR↑ | SSIM↑ | LPIPS↓ | PSNR↑ | SSIM↑ | LPIPS↓ | PSNR↑ | SSIM↑ | LPIPS↓ |
| T-NeRF[46] | 13.163 | 0.709 | 0.353 | 13.818 | 0.739 | 0.324 | 24.944 | 0.742 | 0.336 | 22.242 | 0.700 | 0.363 |
| D-NeRF[46] | 14.158 | 0.697 | 0.352 | 14.660 | 0.737 | 0.312 | 25.380 | 0.766 | 0.300 | 20.791 | 0.692 | 0.349 |
| TiNeuVox[18] | 27.988 | 0.960 | 0.063 | 19.612 | 0.940 | 0.073 | 29.982 | 0.864 | 0.213 | 21.029 | 0.770 | 0.281 |
| T-NeRF$_{PINN}$ | 15.286 | 0.794 | 0.293 | 16.189 | 0.835 | 0.230 | 16.250 | 0.441 | 0.638 | 17.290 | 0.477 | 0.618 |
| HexPlane$_{PINN}$ | 27.042 | 0.958 | 0.057 | 21.419 | 0.946 | 0.067 | 25.215 | 0.763 | 0.389 | 23.091 | 0.742 | 0.401 |
| NSFF[33] | - | - | - | - | - | - | 29.365 | 0.829 | 0.278 | 24.163 | 0.795 | 0.289 |
| **NVFi(Ours)** | **29.027** | **0.970** | **0.039** | **27.594** | **0.972** | **0.036** | **30.675** | **0.877** | **0.211** | **29.745** | **0.876** | **0.204** |

Table 2: Quantitative results of our method and baselines on the NVIDIA Dynamic Scene dataset.

| | Truck | | | | | | Skating | | | | | |
| --- | --- | --- | --- | --- | --- | --- | --- | --- | --- | --- | --- | --- |
| | Interpolation | | | Extrapolation | | | Interpolation | | | Extrapolation | | |
| | PSNR↑ | SSIM↑ | LPIPS↓ | PSNR↑ | SSIM↑ | LPIPS↓ | PSNR↑ | SSIM↑ | LPIPS↓ | PSNR↑ | SSIM↑ | LPIPS↓ |
| TiNeuVox[18] | 27.230 | **0.846** | **0.229** | 24.887 | 0.848 | **0.209** | **29.377** | **0.889** | **0.202** | 24.224 | 0.878 | 0.220 |
| HexPlane$_{PINN}$ | 25.494 | 0.768 | 0.337 | 24.991 | 0.768 | 0.325 | 24.447 | 0.867 | 0.225 | 23.955 | 0.868 | 0.232 |
| **NVFi(Ours)** | **27.276** | 0.840 | 0.235 | **28.269** | **0.855** | 0.220 | 26.999 | 0.848 | 0.227 | **28.654** | **0.896** | **0.208** |

semantic decomposition, the Average Precision (**AP**), Panoptic Quality (**PQ**) and **F1** scores with an IoU threshold of 0.5, together with the mean Intersection over Union (**mIoU**) scores are reported.

## 4.1 Evaluation of Future Frame Extrapolation

We first evaluate the extrapolation capability of our framework. In particular, our method and 5 baselines except NSFF are trained from scratch on each of the 6 objects in Dynamic Object dataset, and all methods on each of the 4 scenes in Dynamic Indoor Scene dataset, all in a scene-specific fashion. In total, $(6 \times 6) + (7 \times 4) = 64$ models are trained for comparison. The keyframe number $K$ is set as 16 in our method for Dynamic Object Dataset and 4 for Dynamic Indoor Scene Dataset. As for real-world NVIDIA Dynamic Scene dataset, we evaluate our model and 2 baselines with comparable performance on our own datasets, where the keyframe number $K$ is set as 4 in our method.

**Analysis:** Table 1 compares all methods regarding the quality of view synthesis for future frame extrapolation. The view synthesis for interpolation is also included for comparison. It can be seen that: 1) our NVFi achieves significantly better results than all baselines on both dynamic datasets, particularly on the critical task of future frame extrapolation, although the strong baseline TiNeuVox shows excellent performance for interpolation. 2) Naïvely adding physics priors into an existing dynamic NeRF tends to be inferior as shown by T-NeRF$_{PINN}$ and HexPlane$_{PINN}$. This clearly verifies the effectiveness of our special design of the joint keyframe and interframe optimization strategy. More qualitative results are in Figure 4(a). Table 2 compares our method and two best baselines. It can be seen that, even if our NVFi only gets comparable performance in interpolation, it still achieves the best performance in extrapolation without any performance drop compared with interpolation thanks to the accurate motion predictions. More qualitative results are in Figure 3.

## 4.2 Evaluation of 3D Semantic Scene Decomposition

Having the well-trained keyframe dynamic radiance field $f_\Theta$ and interframe velocity field $g_\Phi$ for each 3D scene in the Dynamic Indoor Scene dataset in Section 4.1, naturally, all individual 3D objects such as chairs and tables undergoing different movements are supposed to be automatically discovered, segmented, and tracked without needing any extra object annotations as supervision signals.

In order to achieve this desirable unsupervised object decomposition objective, a naïve strategy is to query the velocity values of dense 3D points in space, followed by a velocity clustering module to group points into objects. However, such a strategy fundamentally fails to recognize object shapes but just identifies similar motions, thereby tracking objects is also infeasible. A more elegant strategy is to directly learn an object code $o_p$ (usually one-hot) for each 3D point $p$ within the entire 3D scene volume at the initial timestamp $t = 0$, without retraining any neural layer of the well-trained networks $f_\Theta$ and $g_\Phi$ but just using them. This would clearly allow all dynamic 3D objects in the scene to be segmented and tracked over time. To this end, we simply introduce a simple 4-layer MLP as the

semantic scene decomposition network $h_\Psi$. It takes a 3D point $\boldsymbol{p}$ as input, directly regressing its object code $\boldsymbol{o_p}$, i.e., $\boldsymbol{o_p} = h_\Psi(\boldsymbol{p})$ which is optimized using the following steps:

- First, given the well-trained keyframe dynamic radiance field $f_\Theta$, we uniformly sample dense 3D points at timestamp $t = 0$, obtaining their density values. Only 3D points with sufficiently large densities are kept as valid points for the subsequent steps.
- Second, the valid 3D points are fed into our new object segmentation network $h_\Psi$ (initialized but yet to be trained), obtaining their corresponding object codes.
- Third, the valid points are transported to their correspondences at a random timestamp $t'$ using the well-trained velocity field $g_\Phi$. Motion vectors of these points from 0 to $t'$ are computed.
- Lastly, given per-point motions, we employ the dynamic rigid consistency and spatial smoothness losses proposed in OGC [54] to optimize the object segmentation network $h_\Psi$.

Once the object segmentation network $h_\Psi$ is learned, all dynamic objects at time $t = 0$ are segmented. With the aid of the well-trained velocity field, in any subsequent timestamps, all these identified objects can be naturally tracked. In addition, with the aid of well-trained keyframe dynamic radiance field $f_\Theta$, we simply use the accumulated weights computed in volume rendering to combine point object codes along a given light ray, thus rendering accurate object segmentation 2D masks for any camera poses at any given timestamps. More details of the implementation are in Appendix.

We evaluate the semantic scene decomposition ability on the Dynamic Indoor Scene dataset. In particular, we render all 2D object segmentation masks from our network at the 30 viewing angles over 60 frames for all scenes, i.e., 7200 2D masks, and then evaluate them against ground truth masks. For a fair comparison, we also train a similar object segmentation network for the baseline D-NeRF [46] at time $t = 0$, using its learned deformation vectors as supervision signals and tracking signals. Note that, the deformation vectors are converted back as motion vectors. In addition, we include an image-based object segmentation method, the powerful Mask2Former [11] pre-trained on COCO [34] dataset, as a fully-supervised baseline. More implementation details are in Appendix.

Table 3: Quantitative results of scene decomposition on the Synthetic Indoor Scene dataset.

|  | AP↑ | PQ↑ | F1↑ | mIoU↑ |
|---|---|---|---|---|
| Mask2Former [11] | 65.37 | 73.14 | 78.29 | 64.42 |
| D-NeRF [46] | 57.26 | 46.15 | 59.02 | 46.58 |
| **NVFi(Ours)** | **91.21** | **78.74** | **93.75** | **67.64** |

**Analysis:** As shown in Table 3, we can see that: 1) Our object segmentation performance is superior to D-NeRF [46], essentially because our velocity field learns better scene dynamics than the deformation field, thus enabling the object segmentation network to be better optimized. 2) We also clearly surpass the powerful pre-trained Mask2Former [11] on all metrics. The reasons are two-fold. First, we fundamentally rely on motion patterns rather than appearances to discover objects, thus being able to generalize to unseen object types ("Genome") or scenes ("Factory") better than Mask2Former [11]. Second, our learned object field inherently leverages multi-view consistency, thus allowing the segmentation of partially occluded objects. Figure 4(b) shows qualitative results.

### 4.3 Evaluation of Motion Transfer

We further demonstrate the ability of our model to transfer a well-trained velocity field to another separately trained static scene. All objects in the new scene are expected to undergo the same dynamics as learned in the velocity field. The more accurate the learned velocity field, the more realistic and entertaining the new 3D scene will be, after applying the learned dynamics.

In order to evaluate the performance, we create a new 3D scene, called *Gnome-new*, being similar to the scene *Gnome* in our Dynamic Indoor Scene dataset. We apply the same dynamics of *Gnome* on *Gnome-new*, recording 30*60 = 1800 frames as its ground truth observations. To explicitly show the advantage of our learned disentangled velocity field, we sepa-

Table 4: Quantitative results of motion transfer on Synthetic Indoor Scene dataset.

|  | PSNR↑ | SSIM↑ | LPIPS↓ |
|---|---|---|---|
| D-NeRF [46] | 16.124 | 0.327 | 0.550 |
| **NVFi(Ours)** | 16.178 | 0.334 | 0.551 |

rately train a static TensoRF [7] model for *Gnome-new* only using its frames at time $t = 0$. Note that, any other NeRF variants are also applicable here. We then pick up the well-trained velocity field of *Gnome* in Section 4.1, after which we directly apply the learned velocity field on the newly trained static TensoRF model, rendering 30*60 frames for a comparison with the ground truth images. Similarly, we apply the deformation field learned by D-NeRF in Section 4.1 in the same transferring pipeline, rendering 2D images for comparison. More implementation details are in Appendix.

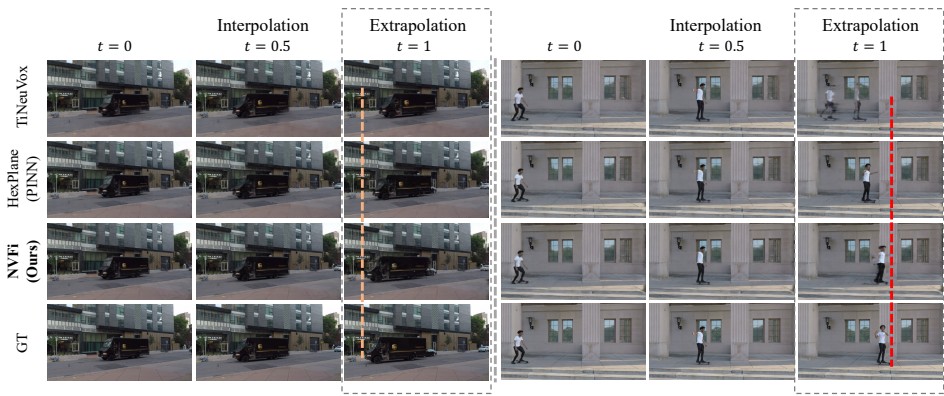

Figure 3: Qualitative results of baselines and our method on NVIDIA Dynamic Scene dataset.

**Analysis:** Figure 4(c) shows that our method clearly keeps the geometry and appearance of the new static object, thanks to the accurately learned disentangled velocity field, whereas D-NeRF [46] fails to do so. However, we observe that in Table 4 our advantage is not so significant. This is because the static reconstruction of *Gnome-new* lacks supervision signals for ground planes occluded by objects, leading to rendering artifacts in these regions when objects are moved away by motion transfer. Our strong ability of motion transfer is further validated by additional experiments on our Dynamic Object Dataset in Appendix.

## 4.4 Ablation Study

**(1) Without Joint Optimization:** We only use $\ell_{keyframe}$ to train the keyframe dynamic radiance field $f_\Theta$, followed by the $(\ell_{divergence\_free} + \ell_{momentum} + \ell_{interframe})$ to separately train the velocity field $g_\Phi$.

**(2) Removing Physics Constraints:** The PINN losses $(\ell_{divergence\_free} + \ell_{momentum})$ are removed to train $g_\Phi$.

**(3) Choice of Keyframe Number $K$:** We set the keyframe number $K$ as 8 and 32, while we choose $K = 16$ in main experiments.

**(4) Reducing the Number of Cameras:** we reduce the number of cameras used in our Dynamic Object datasets to half of the number, *i.e.*, 6 cameras, and one quarter of the number, *i.e.*, 3 cameras.

Table 5 and Table 6 shows the ablation results for future frame extrapolation on our Dynamic Object dataset. We can see that: 1) The joint keyframe and interframe

Table 5: Quantitative results of ablation studies on Dynamic Object dataset.

| | | | Extrapolation | | |
|---|---|---|---|---|---|
| Joint | Physics | #K | PSNR↑ | SSIM↑ | LPIPS↓ |
| ✓ | ✓ | 16 | **27.594** | 0.972 | **0.036** |
| ✗ | ✓ | 16 | 24.792 | 0.955 | 0.059 |
| ✓ | ✗ | 16 | 25.537 | 0.968 | 0.040 |
| ✓ | ✓ | 8 | 27.490 | **0.974** | **0.036** |
| ✓ | ✓ | 32 | 24.902 | 0.964 | 0.037 |

Table 6: Quantitative results of ablation studies on Dynamic Object dataset.

| | Extrapolation | | |
|---|---|---|---|
| Camera Numbers | PSNR↑ | SSIM↑ | LPIPS↓ |
| 12 | **27.594** | **0.972** | **0.036** |
| 6 | 25.114 | 0.959 | 0.122 |
| 3 | 21.370 | 0.917 | 0.084 |

optimization strategy is critical to enable our method to learn accurate velocity field as well as dynamic radiance field. 2) Once the keyframe number $K$ becomes larger, the extrapolation capability clearly drops, validating that the dense supervision is inferior to help learn physics velocity overall. 3) The extremely sparse camera views are unlikely to capture sufficient visual information for physical motion learning. More ablation results are in Appendix.

## 5 Conclusion

In this paper, we extend the appealing radiance field to represent dynamic 3D scenes from multi-view videos. Unlike most of the existing methods which focus on novel view synthesis within the training time period, our method learns to disentangle the physical velocity field from the geometry and appearance of 3D scenes by jointly optimizing two neural networks: the keyframe dynamic radiance field and the interframe velocity field. Extensive experiments on three dynamic datasets demonstrate that our framework learns accurate velocity, enabling successful applications in future frame extrapolation, semantic scene decomposition, and motion transfer.

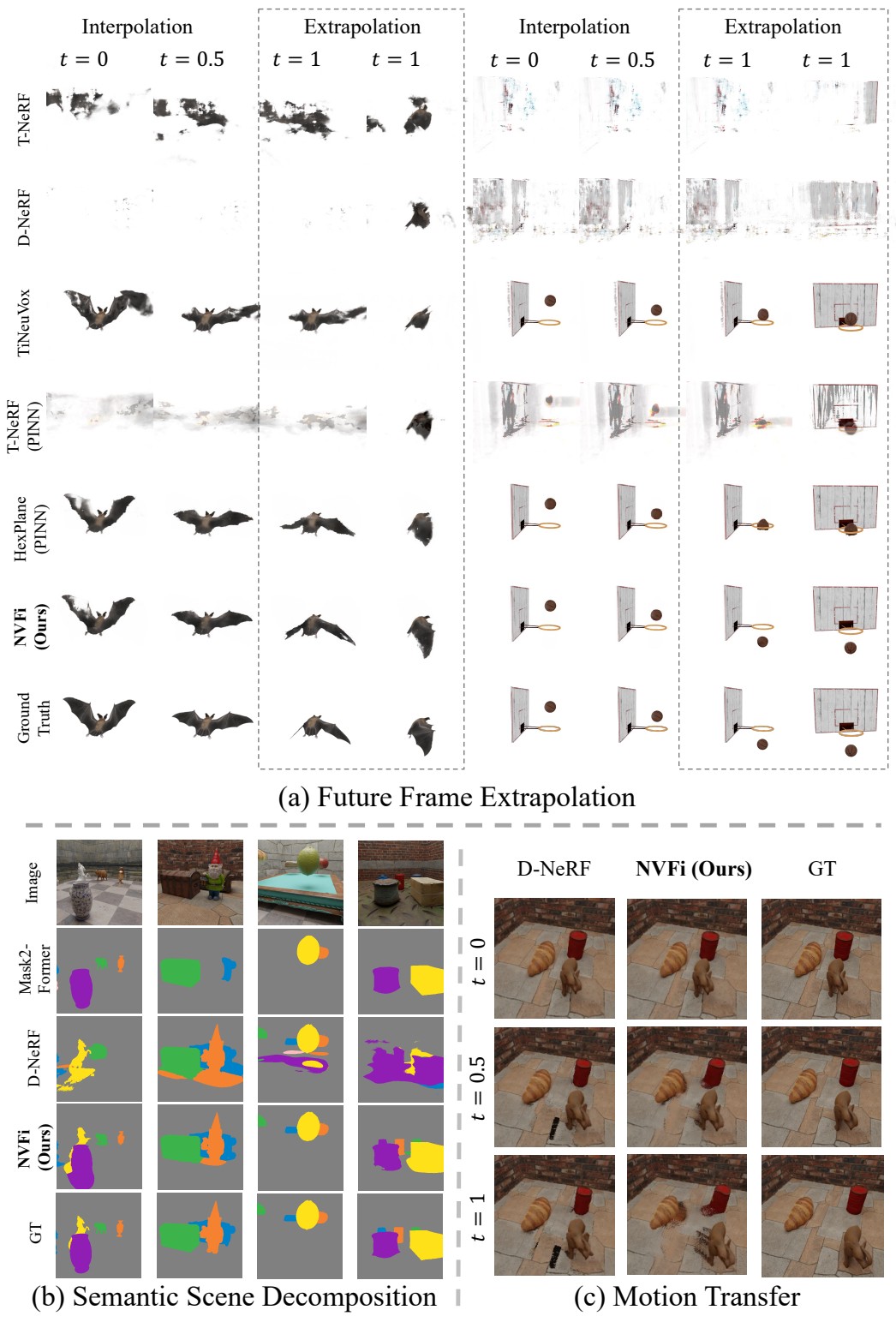

(a) Future Frame Extrapolation

(b) Semantic Scene Decomposition

(c) Motion Transfer

Figure 4: Qualitative results of baselines and our method on the three tasks. More qualitative results can be found in Appendix and our project page: https://vlar-group.github.io/NVFi.html

**Acknowledgements:** This work was supported in part by Research Grants Council of Hong Kong under Grants 25207822 & 15225522.

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

# A  Appendix

The appendix includes:

- Rendering equation.
- Implementation details of keyframe dynamic radiance field and velocity field.
- Additional details of physics prior.
- Implementation details of joint optimization.
- Additional details of datasets.
- Implementation details about semantic scene decomposition.
- Implementation details about motion transfer.
- Additional quantitative results for ablation study.
- Additional quantitative & qualitative results for future frame extrapolation.
- Additional quantitative & qualitative results for semantic scene decomposition.
- Additional qualitative results for motion transfer.

In addition, we provide a **demo video** for better visualization of qualitative results on multiple tasks, appended in the supplementary materials.

## A.1  Rendering Equation

Neural radiance fields were initially introduced [37] to model 3D scenes by associating the coordinate $(x, y, z)$ and view direction $(\theta, \phi)$ of each point in space with its color, represented by vector $\mathbf{c}$, and density, denoted by $\sigma$. We keep the same rendering equation to obtain the expected color $C(\mathbf{r}_i, t_i)$ of a pixel in the image captured by a camera at time $t_i$, a ray $\mathbf{r}_i(s) = \mathbf{o}_i + s\mathbf{d}_i$ is involved, which marches from the camera's center towards the pixel. Here, $\mathbf{o}_i$ and $\mathbf{d}_i$ represent the ray's origin and direction, respectively, while $s$ signifies the distance from a point to the camera, ranging from a predefined near bound $s_n$ to a far bound $s_f$. The pixel color is rendered by sampling a series of points along the ray and performing classical volume rendering:

$$\hat{C}(\mathbf{r}) = \sum_{i=1}^{N} T_i (1 - \exp(-\sigma_i \delta_i)) \mathbf{c}_i,$$
$$T_i = \exp(-\sum_{j=1}^{i-1} \sigma_j \delta_j), \tag{11}$$

where $\delta_i$ represents the distance between the $i_{\text{th}}$ and $(i+1)_{\text{th}}$ sample point, and N denotes the number of sampled points. Equation (11) connects real 3D points with image pixels by accumulating colors $\mathbf{c}_i$ and densities $\sigma_i$ of sample points along the ray.

## A.2  Implementation Details of Keyframe Dynamic Radiance Field

We adapt HexPlane[6] to parameterize volumetric keyframes. It is a direct extension of static TensoRF[7] to a dynamic radiance field. Instead of using concatenation to combine the features from different planes as TensoRF and HexPlane, we use Hadamard Production as used in K-Plane[19], which shows stronger ability in representing complex geometries. The formal definition of our Keyframe Dynamic Radiance Field is as follows:

If we denote $\tau_k$ as the time step corresponding to the $k^{\text{th}}$ keyframe, we can write:

$$A(\mathbf{x}_i, \tau_k) = \mathbf{B}(\mathbf{f}_1(x_i, y_i) \odot \mathbf{g}_1(z_i, \tau_k) \odot \mathbf{f}_2(x_i, z_i) \odot \mathbf{g}_2(y_i, \tau_k) \odot \mathbf{f}_3(y_i, z_i) \odot \mathbf{g}_3(x_i, \tau_k)), \tag{12}$$

and

$$\sigma(\mathbf{x}_i, \tau_k) = \mathbf{1}^{\top}(\mathbf{h}_1(x_i, y_i) \odot \mathbf{k}_1(z_i, \tau_k) \odot \mathbf{h}_2(x_i, z_i) \odot \mathbf{k}_2(y_i, \tau_k) \odot \mathbf{h}_3(y_i, z_i) \odot \mathbf{k}_3(x_i, \tau_k)),$$

where $\sigma(\mathbf{x}_i, \tau_k)$ is the predicted sigma for point $\mathbf{x}_i$, $A(\mathbf{x}_i, \tau_k)$ is the predicted appearance feature. And $\mathbf{f}_j, \mathbf{h}_j, \mathbf{g}_j$ and $\mathbf{k}_j$ are vector-valued grid functions with output dimension $M_\sigma$ or $M_{app}$ respectively. The

final density value $\sigma$ is the summation of the $M_\sigma$ dimension density feature, and the final appearance feature is a weighted sum of the $M_{app}$ dimension output feature, where $\mathbf{B}$ is a simple linear layer. Note unlike HexPlane[6] and K-Planes[19], we only do grid interpolation to find the feature values for spatial coordinates, while we pick the time dimension according to the corresponding keyframe index. The final RGB color $\mathbf{c}$ is regressed from a 2-layer MLP with appearance feature and view direction as inputs. With points' density $\sigma$ and colors $\mathbf{c}$, images are rendered via volumetric rendering Equation (11).

In our implementation, for all the experiments, we set $M_\sigma$ as 24 and $M_{app}$ as 48, while for different experiments, we use different keyframe numbers. For object extrapolation task, we set keyframe number as 16, due to the more complex dynamics of the deformable objects. For both indoor extrapolation task and segmentation task, we set keyframe number as 4. The color decoder MLP has 2 hidden layers, each with 128 nodes.

### A.3 Implementation Details of Velocity Field

The velocity field is defined as follows:

$$\boldsymbol{v} = g_\Phi(\boldsymbol{p}, t) = g_\Phi(x, y, z, t) = w_\Phi(x, y, z, t)\mathbf{M_v}(x, y, z) \quad \Phi \text{ are trainable parameters of MLPs.} \quad (13)$$

Here we decompose the velocity field as a weight MLP $w_\Phi$ along with a velocity basis field. In our experiment, we find this can make the velocity convergence faster. The velocity basis field is defined as:

$$\mathbf{M_v}(x, y, z) = \begin{bmatrix} 1 & 0 & 0 \\ 0 & 1 & 0 \\ 0 & 0 & 1 \\ -z & 0 & x \\ -y & x & 0 \\ 0 & -z & y \end{bmatrix}. \quad (14)$$

The weight MLP is implemented as 4 hidden layers with 128 nodes, with a positional encoder of dim 3 as in [37]. The output is a weight vector $\mathbf{w} \in \mathbb{R}^6$, which could be regarded as the linear velocity in x, y, z directions and angular velocity rounding x, y, z axis respectively.

### A.4 Additional Details of Physics Prior

We enforce the appearance and volume density features, denoted as $f_i$ both admit the conservation laws characterized by the divergence-free implicit velocity field $\mathbf{v}(\mathbf{x}, t)$:

$$\frac{\partial f_i}{\partial t} + \nabla \cdot (\mathbf{v} f_i) = \frac{\partial f_i}{\partial t} + \mathbf{v} \cdot \nabla f_i + f_i \nabla \cdot \mathbf{v} = \frac{Df_i}{Dt} + f_i \nabla \cdot \mathbf{v} = 0, \quad \text{where } \frac{D}{Dt} \text{ is the material derivative.}$$

Intuitively, we assume the objects moved by the velocity field have a property of local rigidity, and the implicit velocity field should be incompressible (or in other words free of sources or sinks of mass). Then we can get the divergence-free constraint $\nabla \cdot \mathbf{v} = 0$.

The momentum of a velocity field is defined to be $\rho \mathbf{v}$, per unit volume. Newton's second law of motion states that momentum is conserved by a mechanical system of masses if no forces act on the system. If $\mathbf{F}(\mathbf{x}, t)$ is the force acting on the velocity, per unit volume, then we immediately have

$$\rho \frac{D\mathbf{v}_i}{Dt} = \mathbf{F},$$

where we assume the implicit velocity field has uniform mass density $\rho$. As a result, the momentum conservation constraint could be derived as:

$$\frac{D\mathbf{v}_i}{Dt} = \frac{\partial \mathbf{v}}{\partial t} + \mathbf{v} \cdot \nabla \mathbf{v} = \frac{\mathbf{F}}{\rho} = \mathbf{a}. \quad (15)$$

### A.5 Implementation Details of Joint Optimization

During training, our keyframe radiance field starts with a space grid size of $64^3$ and increases its resolution in log scale at 2k, 4k, 6k, 8k, 10k iterations till $200^3$. The learning rate for feature planes is 0.02, and the learning rate for color decoding neural network and velocity field is 0.001. All learning

rates are exponentially decayed to $1/10$ at final iteration 30k. We use Adam[29] for optimization with $\beta_1 = 0.9, \beta_2 = 0.99$. We apply Total Variational loss [7, 6, 19] on all feature planes with $\lambda = 0.001$ for spatial axes.

We sample 262144 points uniformly in the space $[-1, 1]^3$ and time $[0, 1]$ for dynamic object datasets, and 1310672 points for dynamic indoor scene datasets every iteration. The jacobians of velocity required is calculated by using autograd from functorch[26]. For all the sampled points, we only evaluate the physics losses at occupied region, where the grid alpha $\alpha = 1 - \exp -\sigma \times 0.01 \geq 0.0001$. We set the loss weight for divergence-free loss as 5, and the weight for momentum conservation loss as 0.1. All scenes are trained for 1.5 hours on a single NVIDIA RTX 3090 GPU respectively.

## A.6 Additional Details of Datasets

**Dynamic Object dataset:** This dataset comprises 6 distinct objects [1], displaying a variety of motion types, including both rigid and deformable movements. We use a total of 15 views, of which 12 are allocated for training and 3 for testing. Each view consists of 60 frames over 1 second; however, only the first 45 frames are used in the training set, with the remaining 15 frames for evaluation. Details of the 6 dynamic objects are:

- **Falling Ball:** A basketball is falling freely through a hoop. The basketball is accelerated by gravity, which should be learned by the model.
- **Telescope:** A telescope is given, whose upper part is rotating while the lower part remains static.
- **Fan:** An antique fan is given. The outer part of the fan is static, while the inner fan is rotating. The embedded motion makes this scene difficult.
- **Bat:** A bat is flapping its wings. Since the wing is extremely thin, and the motion is in a great extent, it's hard to reconstruct the motion.
- **Shark:** A shark is shaking its tail left and right.
- **Whale:** A whale is flapping its tail up and down.

**Dynamic Indoor Scene dataset:** This dataset consists of 4 indoor scenes, each containing several rigid objects [2] undergoing different rigid body motions. Since the indoor scene is more challenging, it comprises 30 views, with 25 for training and 5 for testing. Like the first dataset, we use the first 45 frames from the 60-frame views for training and reserved the remaining frames for evaluation. A set of ground-truth segmentation masks is provided for evaluation. The segmentation mask is rendered by blender object index, which is perfectly accurate as ground truth. We have in total 4 scenes:

- **Gnome House:** 3 objects are moving outwards from the center of a house: a gnome, a sofa, and a treasure chest.
- **Chessboard:** 5 objects are moving towards each other on a chessboard: a horse statue, a marble bust, and a china vase from one side, a wooden elephant and a brass vase from the other side.
- **Factory:** 5 objects are moving towards nearly the same direction, among which 2 objects are rotating at the same time: three barrels with different appearance, a cardboard box, and a compost bag.
- **Dining Table:** 4 objects are falling down onto a dining table from the surrounding air: an apple, a cake, a croissant, and a lime.

## A.7 Implementation Details about Semantic Scene Decomposition

**NVFi(Ours):** Figure 5 illustrates the optimization pipeline of our method for scene decomposition task:

---

[1]Basketball is downloaded from TurboSquid, licensed under the TurboSquid 3D Model License: https://blog.turbosquid.com/turbosquid-3d-model-license. Other objects are purchased from SketchFab, licensed under the SketchFab Standard License: https://sketchfab.com/licenses

[2]All objects and textures are freely downloaded from Poly Haven, licensed under Poly Haven asset License: https://polyhaven.com/license

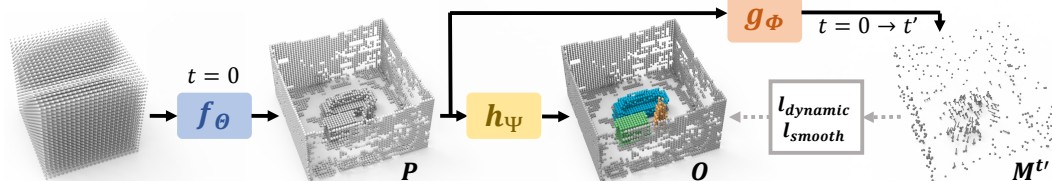

Figure 5: The optimization pipeline of our method for scene decomposition. The solid black lines represent forward inference while the dashed gray lines represent backward supervision.

- First, we sample dense 3D points at timestamp $t = 0$ and query their density values through the well-trained keyframe dynamic radiance field $f_\Theta$. By removing 3D points with low densities, we obtain a non-empty point set $\boldsymbol{P}$ with $N$ points at timestamp $t = 0$, *i.e.*, $\boldsymbol{P} \in \mathbb{R}^{N \times 3}$.
- Second, we feed the point set $\boldsymbol{P}$ into the to-be-trained scene decomposition network $h_\Psi$ and obtain per-point object codes $\boldsymbol{O} \in (0, 1)^{N \times K}$, where $K$ is a predefined number of objects that the network is expected to predict in maximum.
- Third, the points in $\boldsymbol{P}$ are transported to their correspondences $\boldsymbol{P}^{t'}$ using the well-trained velocity field $g_\Phi$. Thus, we can compute the per-point motion vectors $\boldsymbol{M}^{t'} \in \mathbb{R}^{N \times 3}$ as $\boldsymbol{M}^{t'} = \boldsymbol{P}^{t'} - \boldsymbol{P}$.
- Lastly, we apply two losses proposed in OGC [54] onto the predicted object codes. **1) Dynamic rigid consistency:** For the $k^{th}$ object, We first retrieve its (soft) binary mask $\boldsymbol{O}^k$, and feed the tuple $\{\boldsymbol{P}, \boldsymbol{P}^{t'}, \boldsymbol{O}^k\}$ into the weighted-Kabsch algorithm to estimate its transformation matrix $\boldsymbol{T}_k \in \mathbb{R}^{4 \times 4}$ belonging to $SE(3)$ group. Then the dynamic loss is defined as:

$$\ell_{dynamic} = \frac{1}{N} \sum_{\boldsymbol{p} \in \boldsymbol{P}} \left\| \left( \sum_{k=1}^{K} o_{\boldsymbol{p}}^k \cdot (\boldsymbol{T}_k \circ \boldsymbol{p}) \right) - (\boldsymbol{p} + \boldsymbol{m}^{t'}) \right\|_2 \tag{16}$$

where $o_{\boldsymbol{p}}^k$ represents the probability of being assigned to the $k^{th}$ object for a specific point $\boldsymbol{p}$, and $\boldsymbol{m}^{t'} \in \mathbb{R}^3$ represents the motion vector of $\boldsymbol{p}$ from timestamp 0 to $t'$. The operation $\circ$ applies the rigid transformation to the point. This loss serves to discriminate objects with different motions. **2) Spatial smoothness:** We first search $H$ nearest neighboring points for each point $\boldsymbol{p}$ in $\boldsymbol{P}$. Then the smoothness loss is defined as:

$$\ell_{smooth} = \frac{1}{N} \sum_{\boldsymbol{p} \in \boldsymbol{P}} \left( \frac{1}{H} \sum_{h=1}^{H} \| \boldsymbol{o}_{\boldsymbol{p}} - \boldsymbol{o}_{\boldsymbol{p}_h} \|_1 \right) \tag{17}$$

where $\boldsymbol{o}_{\boldsymbol{p}} \in (0, 1)^K$ represents the object assignment of center point $\boldsymbol{p}$, and $\boldsymbol{o}_{\boldsymbol{p}_h} \in (0, 1)^K$ represents the object assignment of its $h^{th}$ neighbouring point. This loss servers to avoid the over-segmentation issues. We refer readers to [54] for more details.

For our scene decomposition network $h_\Psi$, we adopt a simple 4-layer MLP with 128 neurons in each hidden layer. The maximum number of predicted objects $K$ is set to 8. A softmax activation is applied onto the predicted object code. During optimization, we adopt the Adam optimizer with a learning rate of 0.005 and optimize the network for 1000 iterations. The two losses $\ell_{dynamic}$ and $\ell_{smooth}$ are weighted by $\{1.0, 0.1\}$.

**D-NeRF:** For D-NeRF, we adopt exactly the same architecture of scene decomposition network as ours. The optimizer and other hyperparameters are also consistent with ours.

**Mask2Former:** We use the implementation of Mask2Former from MMDetection [3], where we choose the most powerful Swin-L backbone. The model is pretrained on COCO dataset for panoptic segmentation task. Since we aim to decompose objects with different motions only, the ground plane and the wall in each scene are regarded as a whole static background, while the pre-trained model may separate them into different parts. For a fair comparison, we manually merge the segments in model's predictions corresponding to the static background.

---

[3]https://github.com/open-mmlab/mmdetection

Table 7: Quantitative results of ablation study for both interpolation task and extrapolation task on Dynamic Object dataset.

| Joint | Physics | #$K$ | Interpolation | | | Extrapolation | | |
|:-:|:-:|:-:|:-:|:-:|:-:|:-:|:-:|:-:|
| | | | PSNR↑ | SSIM↑ | LPIPS↓ | PSNR↑ | SSIM↑ | LPIPS↓ |
| ✓ | ✓ | 16 | 29.027 | **0.970** | **0.039** | **27.594** | 0.972 | **0.036** |
| ✗ | ✓ | 16 | 27.525 | 0.959 | 0.057 | 24.792 | 0.955 | 0.059 |
| ✓ | ✗ | 16 | 29.109 | **0.970** | **0.039** | 25.537 | 0.968 | 0.040 |
| ✓ | ✓ | 8 | **29.355** | **0.970** | 0.041 | 27.490 | **0.974** | **0.036** |
| ✓ | ✓ | 32 | 28.922 | 0.969 | 0.042 | 24.902 | 0.964 | 0.037 |

Table 8: Quantitative results of ablation study of the keyframe number on our Dynamic Indoor Scene dataset.

| Number of Keyframes | Interpolation | | | Extrapolation | | |
|:-:|:-:|:-:|:-:|:-:|:-:|:-:|
| | PSNR↑ | SSIM↑ | LPIPS↓ | PSNR↑ | SSIM↑ | LPIPS↓ |
| 4 | **30.675** | **0.877** | **0.211** | **29.745** | **0.876** | **0.204** |
| 8 | 30.321 | 0.871 | 0.220 | 29.093 | 0.873 | 0.225 |
| 16 | 28.000 | 0.862 | 0.226 | 26.235 | 0.839 | 0.237 |
| 32 | 29.764 | 0.851 | 0.255 | 26.634 | 0.828 | 0.247 |

## A.8 Additional Implementation Details about Motion Transfer

There are two models to train for this task. The first one is the velocity field model. The training detail of the velocity field is the same as future frame extrapolation task. In fact, we use the trained velocity from our future frame extrapolation task. The other model to train is a new static TensoRF [7] for the scene to be transferred. To be simple, we just set the keyframe number of our NVFi as 1, then we can get a static representation as TensoRF. The next step is to load the parameter from the pretrained velocity field to our new NVFi.

## A.9 Additional Quantitative Results for Ablation Study

Here we show the total results for ablation study in Tables 7&8&9, both for interpolation and extrapolation.

## A.10 Additional Quantitative Results for Future Frame Extrapolation

We report the quantitative results of interpolation and extrapolation tasks for individual objects/scenes in Dynamic Object dataset and Dynamic Indoor Scene dataset. As shown in Tables 10&11, our method achieves leading performance frequently on interpolation, and significantly outperforms all baselines on future frame extrapolation.

## A.11 Additional Quantitative Results for Semantic Scene Decomposition

In the main paper, we report the excellent results of our model in Table 3 on semantic scene decomposition task based on a small integral time step 0.06. We also report the results with much larger integral time step 0.5. As shown in Tables 12&13, when we set the integral time step as 0.06, our segmentation results are significantly boosted, surpassing the two baselines by large margins.

## A.12 Additional Qualitative results for Future Frame Extrapolation

We show the remaining qualitative results for interpolation and extrapolation tasks on Dynamic Objects Datasets in Figures 6&7, on Dynamic Indoor Scene Datasets in Figures 8&9. Images at time 0 and time 0.5 are within the observed time (interpolation), while time 1 is future frames. For each scene, the first three images are novel views, and the fourth one is from a seen viewing angle but at a future timestamp.

## A.13 Additional Qualitative results for semantic scene decomposition

We show more qualitative results for semantic scene decomposition on Dynamic Indoor Scene dataset in Figures 10&11. Images at time 0 till time 0.67 are within observed time interval, while time 0.83 and time 1 is for future frames. Besides, we apply the semantic decomposition pipeline onto the Dynamic Object dataset for part decomposition. Qualitative results are shown in Figure 12.

Table 9: Quantitative results of ablation study of the camera number on our Dynamic Objects dataset.

| Number of Cameras | Interpolation | | | Extrapolation | | |
|---|---|---|---|---|---|---|
| | PSNR↑ | SSIM↑ | LPIPS↓ | PSNR↑ | SSIM↑ | LPIPS↓ |
| 12 | **29.027** | **0.970** | **0.039** | **27.594** | **0.972** | **0.036** |
| 6 | 25.689 | 0.954 | 0.051 | 25.114 | 0.959 | 0.122 |
| 3 | 21.460 | 0.912 | 0.088 | 21.370 | 0.917 | 0.084 |

Table 10: Per-scene quantitative results of Dynamic Object dataset.

| Methods | Falling Ball | | | | | | Bat | | | | | |
|---|---|---|---|---|---|---|---|---|---|---|---|---|
| | Interpolation | | | Extrapolation | | | Interpolation | | | Extrapolation | | |
| | PSNR↑ | SSIM↑ | LPIPS↓ | PSNR↑ | SSIM↑ | LPIPS↓ | PSNR↑ | SSIM↑ | LPIPS↓ | PSNR↑ | SSIM↑ | LPIPS↓ |
| T-NeRF[46] | 14.921 | 0.782 | 0.326 | 15.418 | 0.793 | 0.308 | 13.070 | 0.836 | 0.234 | 13.897 | 0.834 | 0.230 |
| D-NeRF[46] | 15.548 | 0.665 | 0.435 | 15.116 | 0.644 | 0.427 | 14.087 | 0.845 | 0.212 | 15.406 | 0.887 | 0.175 |
| TiNeuVox[18] | 35.458 | 0.974 | 0.052 | 20.242 | 0.959 | 0.067 | 16.080 | 0.908 | 0.108 | 16.952 | 0.930 | 0.115 |
| T-NeRF$_{PINN}$ | 17.687 | 0.775 | 0.368 | 17.857 | 0.829 | 0.265 | 16.412 | 0.903 | 0.197 | 18.983 | 0.930 | 0.132 |
| HexPlane$_{PINN}$ | 32.144 | 0.965 | 0.065 | 20.762 | 0.951 | 0.081 | **23.399** | 0.958 | 0.057 | 21.144 | 0.951 | 0.064 |
| **NVFi(Ours)** | **35.826** | **0.978** | **0.041** | **31.369** | **0.978** | **0.041** | 23.325 | **0.964** | **0.046** | **25.015** | **0.968** | **0.042** |

| Methods | Fan | | | | | | Telescope | | | | | |
|---|---|---|---|---|---|---|---|---|---|---|---|---|
| | Interpolation | | | Extrapolation | | | Interpolation | | | Extrapolation | | |
| | PSNR↑ | SSIM↑ | LPIPS↓ | PSNR↑ | SSIM↑ | LPIPS↓ | PSNR↑ | SSIM↑ | LPIPS↓ | PSNR↑ | SSIM↑ | LPIPS↓ |
| T-NeRF[46] | 8.001 | 0.308 | 0.646 | 8.494 | 0.392 | 0.593 | 13.031 | 0.615 | 0.472 | 13.892 | 0.670 | 0.417 |
| D-NeRF[46] | 7.915 | 0.262 | 0.690 | 8.624 | 0.370 | 0.623 | 13.295 | 0.609 | 0.469 | 14.967 | 0.700 | 0.385 |
| TiNeuVox[18] | 24.088 | 0.930 | 0.104 | 20.932 | 0.935 | 0..078 | **31.666** | **0.982** | **0.041** | 20.456 | 0.921 | 0.067 |
| T-NeRF$_{PINN}$ | 9.233 | 0.541 | 0.508 | 9.828 | 0.606 | 0.443 | 14.293 | 0.739 | 0.366 | 15.752 | 0.804 | 0.298 |
| HexPlane$_{PINN}$ | 22.822 | 0.921 | 0.079 | 19.724 | 0.919 | 0.080 | 25.381 | 0.948 | 0.066 | 23.165 | 0.932 | 0.074 |
| **NVFi(Ours)** | **25.213** | **0.948** | **0.049** | **27.172** | **0.963** | **0.037** | 26.487 | 0.959 | 0.048 | **27.101** | **0.963** | **0.046** |

| Methods | Shark | | | | | | Whale | | | | | |
|---|---|---|---|---|---|---|---|---|---|---|---|---|
| | Interpolation | | | Extrapolation | | | Interpolation | | | Extrapolation | | |
| | PSNR↑ | SSIM↑ | LPIPS↓ | PSNR↑ | SSIM↑ | LPIPS↓ | PSNR↑ | SSIM↑ | LPIPS↓ | PSNR↑ | SSIM↑ | LPIPS↓ |
| T-NeRF[46] | 13.813 | 0.853 | 0.223 | 15.325 | 0.882 | 0.193 | 16.141 | 0.860 | 0.212 | 15.880 | 0.860 | 0.203 |
| D-NeRF[46] | 17.727 | 0.903 | 0.150 | 19.078 | 0.936 | 0.092 | 16.373 | 0.898 | 0.154 | 14.771 | 0.883 | 0.171 |
| TiNeuVox[18] | 23.178 | 0.971 | 0.059 | 19.463 | 0.950 | 0.050 | **37.455** | **0.994** | **0.016** | 19.624 | 0.943 | 0.063 |
| T-NeRF$_{PINN}$ | 17.315 | 0.878 | 0.177 | 18.739 | 0.921 | 0.115 | 16.778 | 0.927 | 0.141 | 15.974 | 0.919 | 0.127 |
| HexPlane$_{PINN}$ | 28.874 | 0.976 | 0.040 | 22.330 | 0.961 | 0.047 | 29.634 | 0.981 | 0.035 | 21.391 | 0.961 | 0.053 |
| **NVFi(Ours)** | **32.072** | **0.984** | **0.024** | **28.874** | **0.982** | **0.021** | 31.240 | 0.986 | 0.025 | **26.032** | **0.978** | **0.029** |

## A.14 Additional Qualitative Results for Motion Transfer

We have already shown results for Dynamic Indoor Scene datasets in Section 4.3. Here we report three results on objectwise motion transfer in Figure 13:

- **Whale to starfish.** We transfer the flapping tail motion to the starfish. The starfish is supposed to flapping its rays as the whale.

- **Shark to whale.** We transfer the shaking tail motion to the whale. The whale used to flap its tail, but now it's supposed to shake its tail.

- **Whale to shark.** We transfer the flapping tail motion to the shark. The shark is supposed to flap its tail as a whale instead of shaking it.

Table 11: Per-scene quantitative results of Dynamic Indoor Scene dataset.

| Methods | Gnome House | | | | | | Chessboard | | | | | |
|---|---|---|---|---|---|---|---|---|---|---|---|---|
| | Interpolation | | | Extrapolation | | | Interpolation | | | Extrapolation | | |
| | PSNR↑ | SSIM↑ | LPIPS↓ | PSNR↑ | SSIM↑ | LPIPS↓ | PSNR↑ | SSIM↑ | LPIPS↓ | PSNR↑ | SSIM↑ | LPIPS↓ |
| T-NeRF[46] | 26.094 | 0.716 | 0.383 | 23.485 | 0.643 | 0.419 | 25.517 | 0.796 | 0.294 | 20.228 | 0.708 | 0.365 |
| D-NeRF[46] | 27.000 | 0.745 | 0.319 | 21.714 | 0.641 | 0.367 | 24.852 | 0.774 | 0.308 | 19.455 | 0.675 | 0.384 |
| TiNeuVox[18] | 30.646 | **0.831** | **0.253** | 21.418 | 0.699 | 0.326 | **33.001** | **0.917** | **0.177** | 19.718 | 0.765 | 0.310 |
| T-NeRF$_{PINN}$ | 15.008 | 0.375 | 0.668 | 16.200 | 0.409 | 0.651 | 16.549 | 0.457 | 0.621 | 17.197 | 0.472 | 0.618 |
| HexPlane$_{PINN}$ | 23.764 | 0.658 | 0.510 | 22.867 | 0.658 | 0.510 | 24.605 | 0.778 | 0.412 | 21.518 | 0.748 | 0.428 |
| NSFF[33] | **31.418** | 0.821 | 0.294 | 25.892 | 0.750 | 0.327 | 32.514 | 0.810 | 0.201 | 21.501 | 0.805 | 0.282 |
| **NVFi(Ours)** | 30.667 | 0.824 | 0.277 | **30.408** | **0.826** | **0.273** | 30.394 | 0.888 | 0.215 | **27.840** | **0.872** | **0.219** |
| | Factory | | | | | | Dining Table | | | | | |
| Methods | Interpolation | | | Extrapolation | | | Interpolation | | | Extrapolation | | |
| | PSNR↑ | SSIM↑ | LPIPS↓ | PSNR↑ | SSIM↑ | LPIPS↓ | PSNR↑ | SSIM↑ | LPIPS↓ | PSNR↑ | SSIM↑ | LPIPS↓ |
| T-NeRF[46] | 26.467 | 0.741 | 0.328 | 24.276 | 0.722 | 0.344 | 21.699 | 0.716 | 0.338 | 20.977 | 0.725 | 0.324 |
| D-NeRF[46] | 28.818 | 0.818 | 0.252 | 22.959 | 0.746 | 0.303 | 20.851 | 0.725 | 0.319 | 19.035 | 0.705 | 0.341 |
| TiNeuVox[18] | 32.684 | 0.909 | **0.148** | 22.622 | 0.810 | 0.229 | 23.596 | 0.798 | 0.274 | 20.357 | 0.804 | 0.258 |
| T-NeRF$_{PINN}$ | 16.634 | 0.446 | 0.624 | 17.546 | 0.480 | 0.609 | 16.807 | 0.486 | 0.640 | 18.215 | 0.548 | 0.595 |
| HexPlane$_{PINN}$ | 27.200 | 0.826 | 0.283 | 24.998 | 0.792 | 0.312 | 25.291 | 0.788 | 0.350 | 22.979 | 0.771 | 0.355 |
| NSFF[33] | **33.975** | **0.919** | 0.152 | 26.647 | 0.855 | 0.196 | 19.552 | 0.665 | 0.464 | 22.612 | 0.770 | 0.351 |
| **NVFi(Ours)** | 32.460 | 0.912 | 0.151 | **31.719** | **0.908** | **0.154** | **29.179** | **0.885** | **0.199** | **29.011** | **0.898** | **0.171** |

Table 12: Overall quantitative results for semantic scene decomposition.

| | AP↑ | PQ↑ | F1↑ | Pre↑ | Rec↑ | mIoU↑ |
|---|---|---|---|---|---|---|
| Mask2Former[11] | 65.37 | 73.14 | 78.29 | **94.83** | 68.88 | 64.42 |
| D-NeRF[46] | 57.26 | 46.15 | 59.02 | 56.55 | 62.94 | 46.58 |
| **NVFi(Ours, $\Delta t = 0.5$)** | 75.82 | 63.34 | 80.59 | 80.78 | 81.11 | 57.05 |
| **NVFi(Ours, $\Delta t = 0.06$)** | **91.21** | **78.74** | **93.75** | 93.76 | **93.74** | **67.64** |

Table 13: Per-scene quantitative results for semantic scene decomposition.

| Methods | Gnome House | | | | | | Chessboard | | | | | |
|---|---|---|---|---|---|---|---|---|---|---|---|---|
| | AP↑ | PQ↑ | F1↑ | Pre↑ | Rec↑ | mIoU↑ | AP↑ | PQ↑ | F1↑ | Pre↑ | Rec↑ | mIoU↑ |
| Mask2Former[11] | 60.89 | 73.05 | 77.32 | 85.32 | 70.69 | 66.94 | 82.68 | 81.35 | 90.81 | 97.54 | 84.94 | **76.17** |
| D-NeRF[46] | 80.54 | 62.24 | 85.28 | 85.28 | 85.28 | 54.82 | 57.12 | 48.11 | 60.22 | 56.20 | 64.85 | 48.97 |
| **NVFi(Ours, $\Delta t = 0.5$)** | 99.00 | 82.99 | 99.92 | 99.92 | 99.92 | 66.42 | 48.30 | 43.14 | 61.19 | 61.53 | 60.84 | 45.95 |
| **NVFi(Ours, $\Delta t = 0.06$)** | **100.00** | **85.01** | **100.00** | **100.00** | **100.00** | **68.01** | 67.97 | 57.95 | 76.96 | 76.96 | 74.96 | 56.79 |
| Methods | Factory | | | | | | Dining Table | | | | | |
| | AP↑ | PQ↑ | F1↑ | Pre↑ | Rec↑ | mIoU↑ | AP↑ | PQ↑ | F1↑ | Pre↑ | Rec↑ | mIoU↑ |
| Mask2Former[11] | 40.25 | 53.54 | 57.60 | 99.01 | 40.61 | 37.76 | 77.65 | 84.61 | 87.42 | 97.44 | 79.28 | **76.80** |
| D-NeRF[46] | 17.33 | 17.08 | 21.29 | 25.35 | 18.35 | 20.72 | 74.05 | 57.15 | 69.30 | 59.35 | 83.27 | 61.82 |
| **NVFi(Ours, $\Delta t = 0.5$)** | 64.82 | 57.75 | 77.93 | 85.72 | 71.44 | 49.70 | 91.17 | 69.49 | 83.31 | 75.96 | 92.23 | 66.13 |
| **NVFi(Ours, $\Delta t = 0.06$)** | **98.86** | **80.17** | **99.09** | **99.09** | **99.09** | **69.07** | **98.01** | **91.81** | **98.95** | **98.99** | **98.92** | 76.68 |

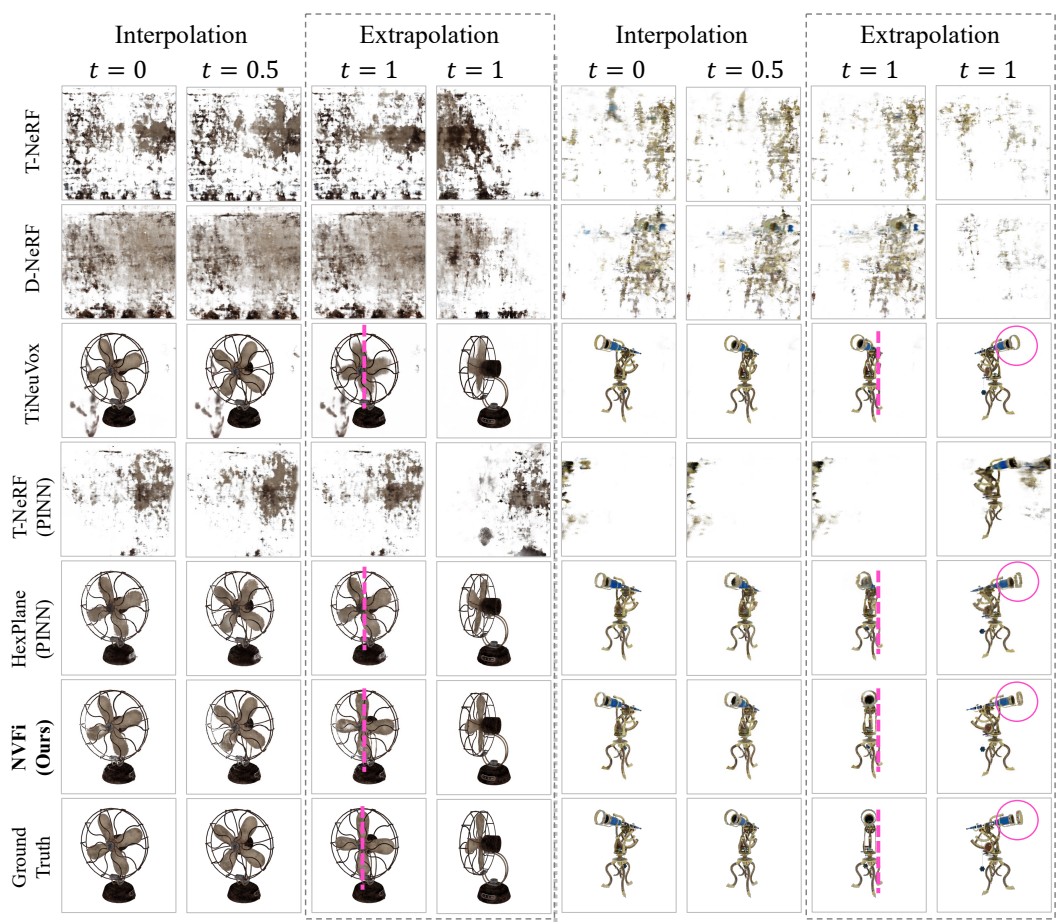

Figure 6: Qualitative results of objects Fan and Telescope.

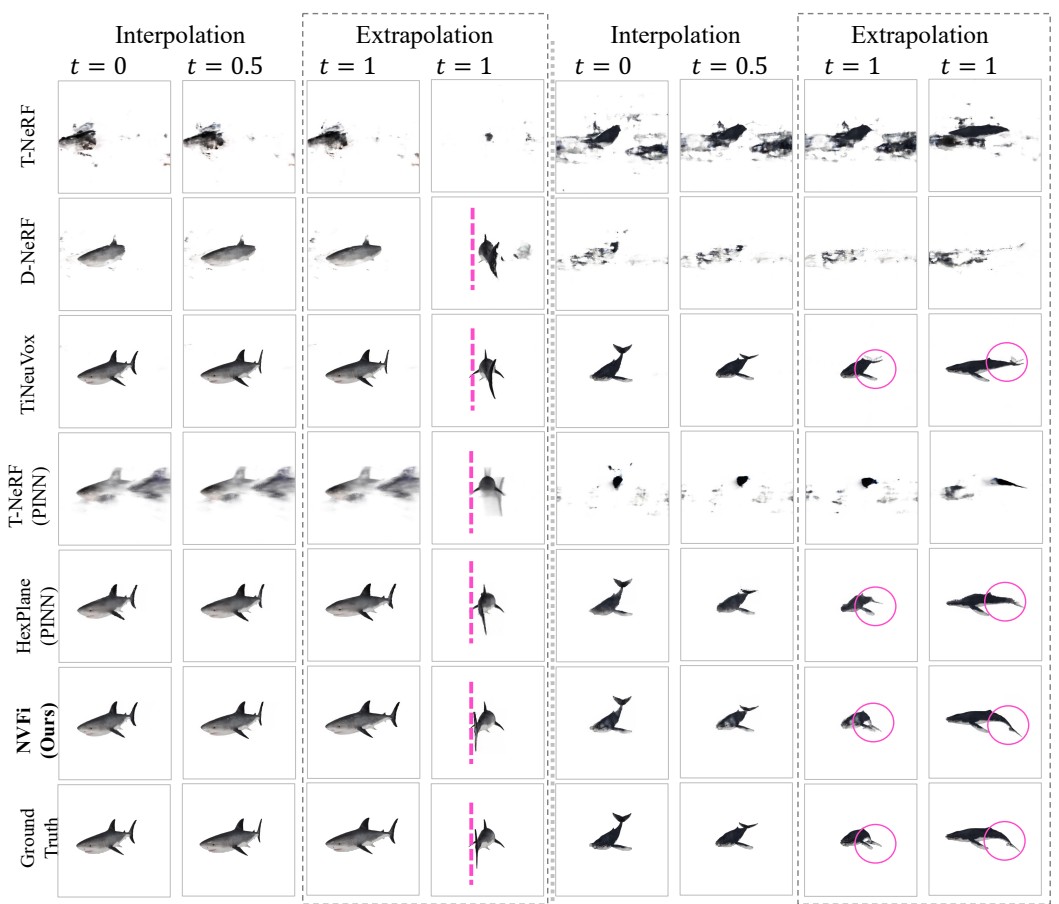

Figure 7: Qualitative results of objects Shark and Whale.

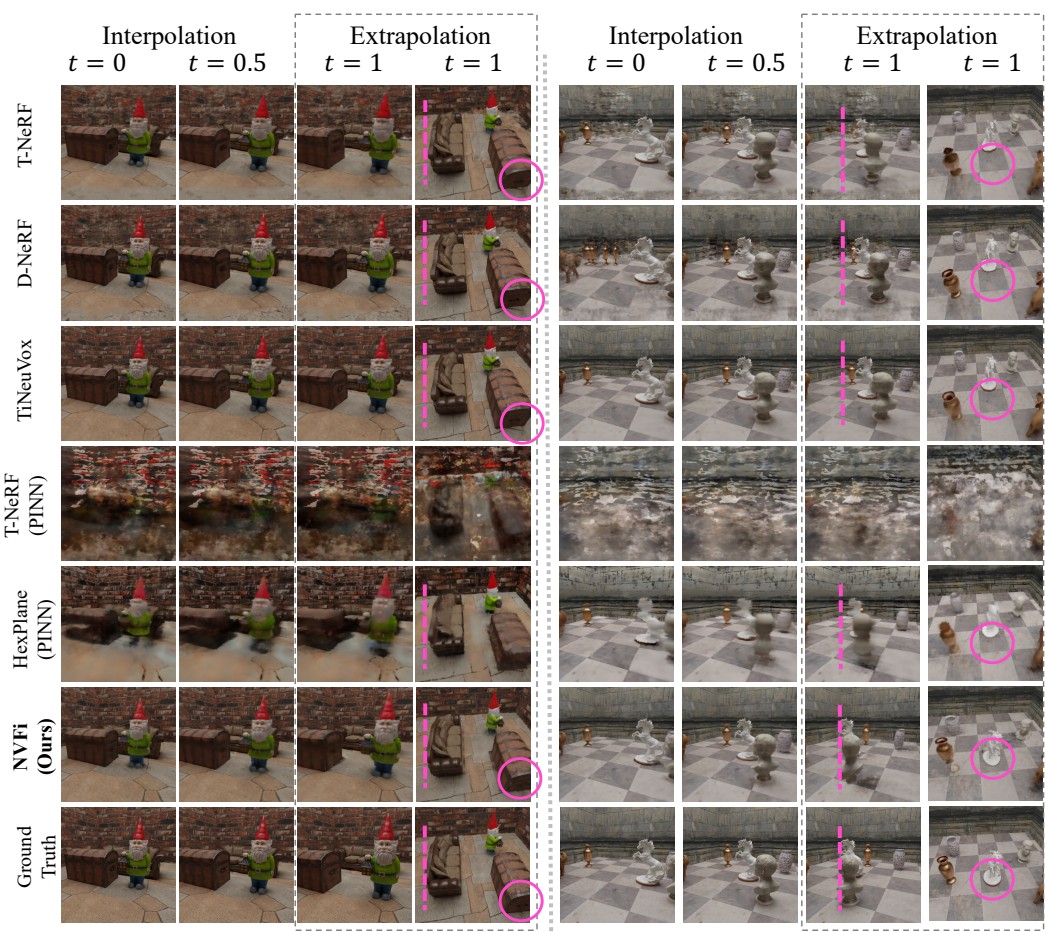

Figure 8: Qualitative results of scenes Gnome House and Chessboard.

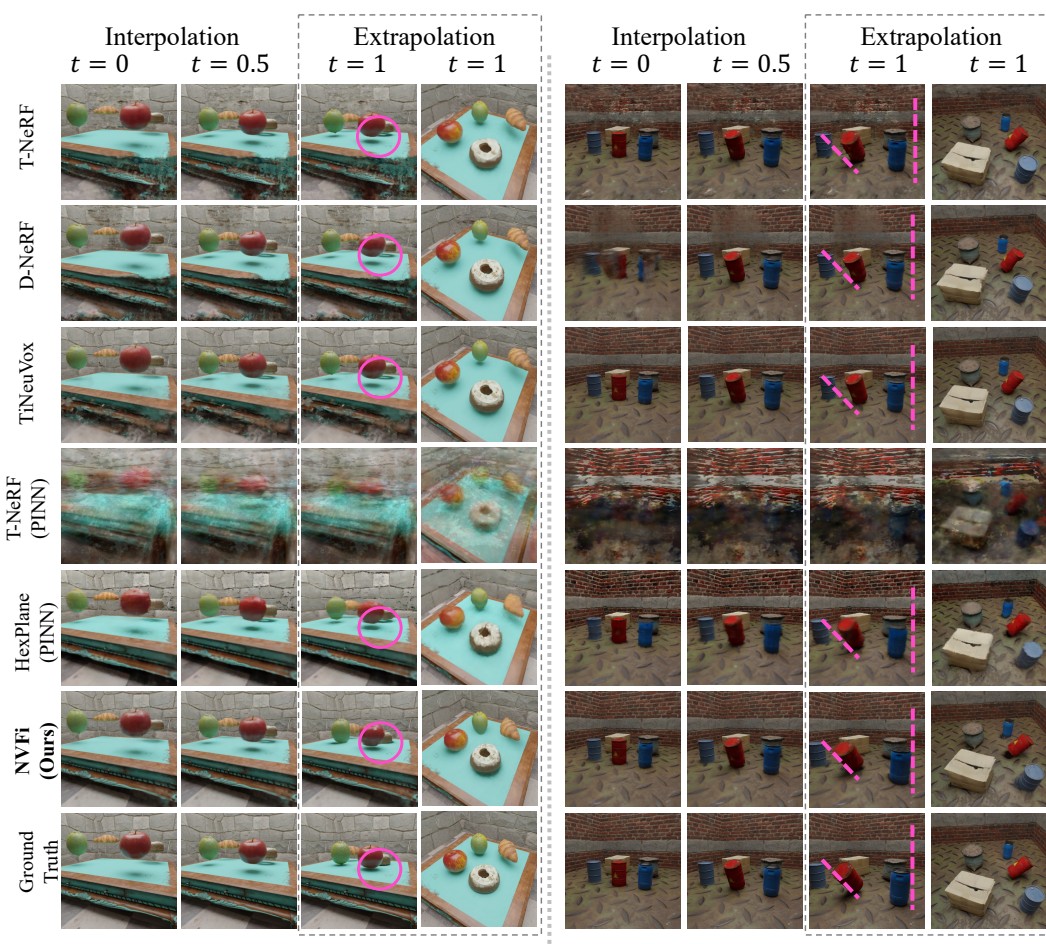

Figure 9: Qualitative results of scenes Dining Table and Factory.

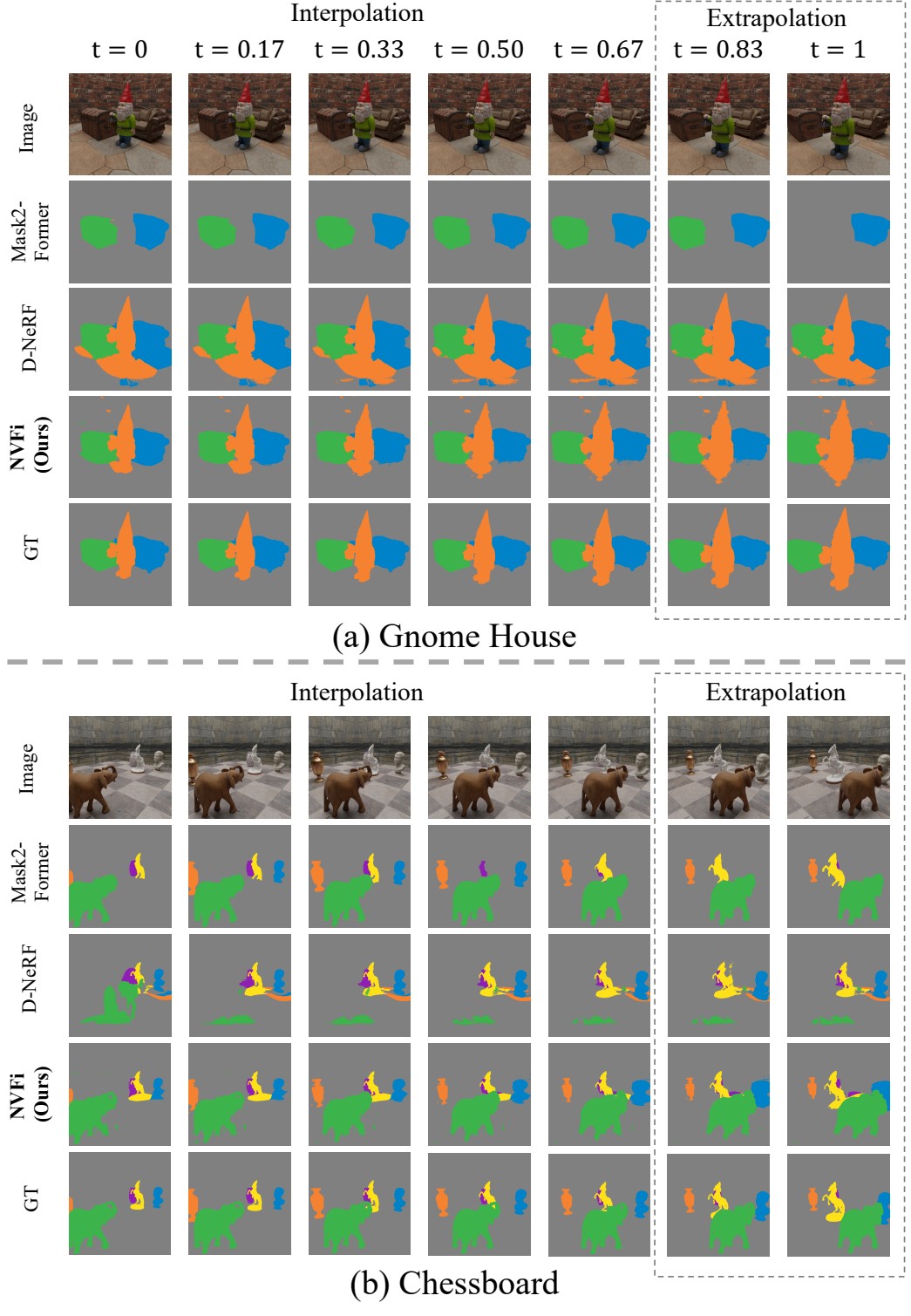

Figure 10: Qualitative results of semantic scene decomposition for Gnome House and Chessboard.

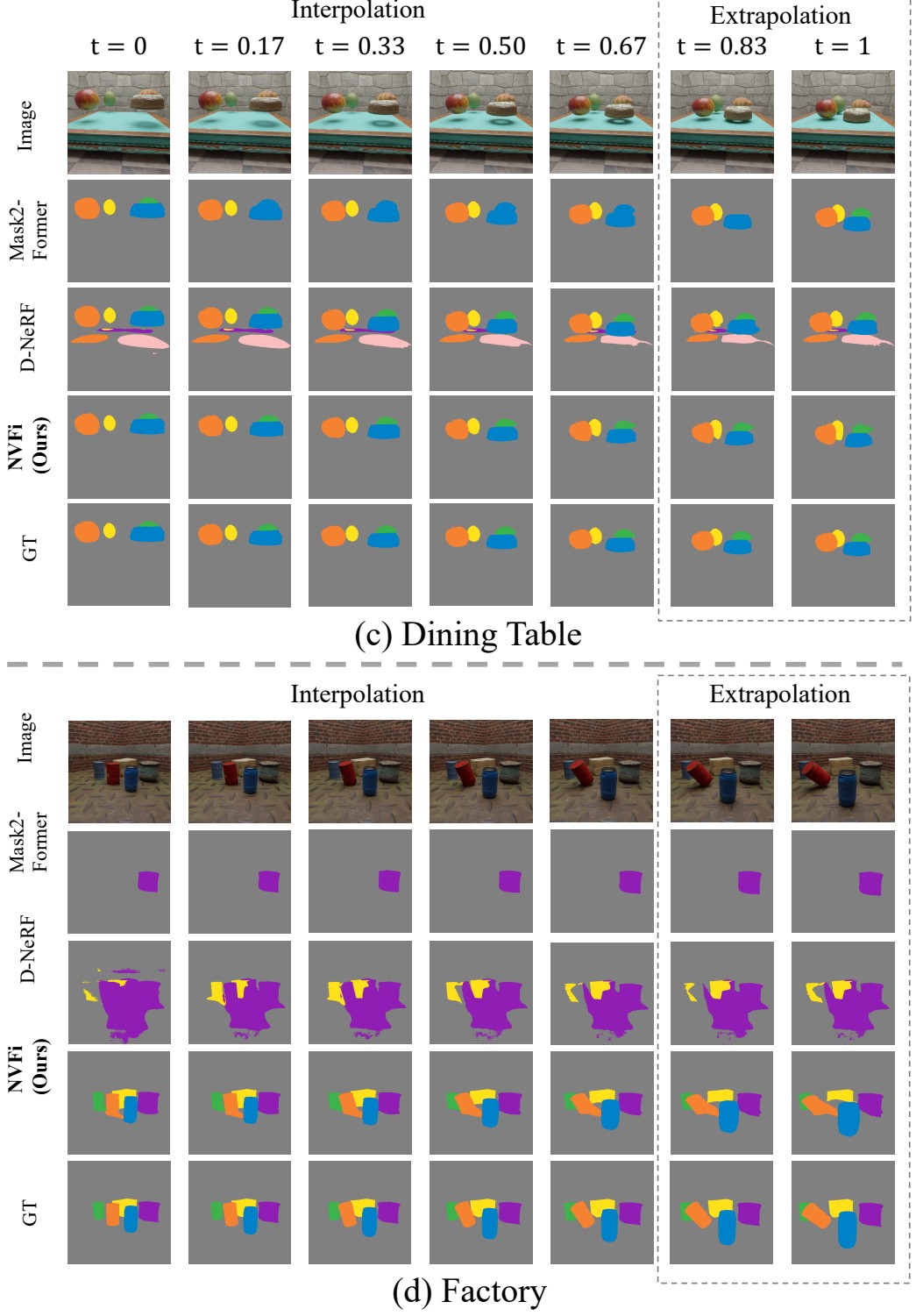

Figure 11: Qualitative results of semantic scene decomposition for Dining Table and Factory.

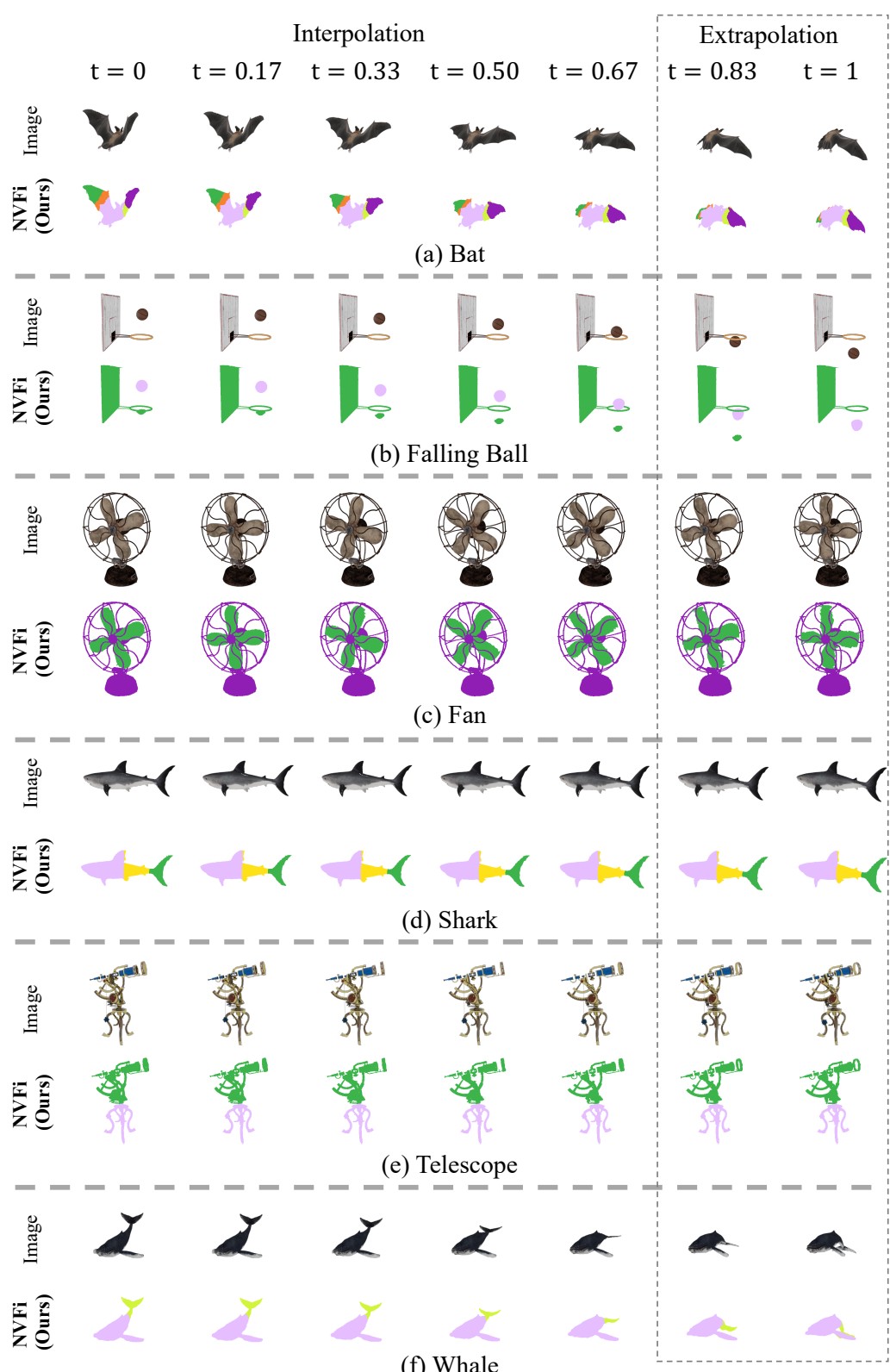

Figure 12: Qualitative results of semantic scene decomposition for Dynamic Object dataset.

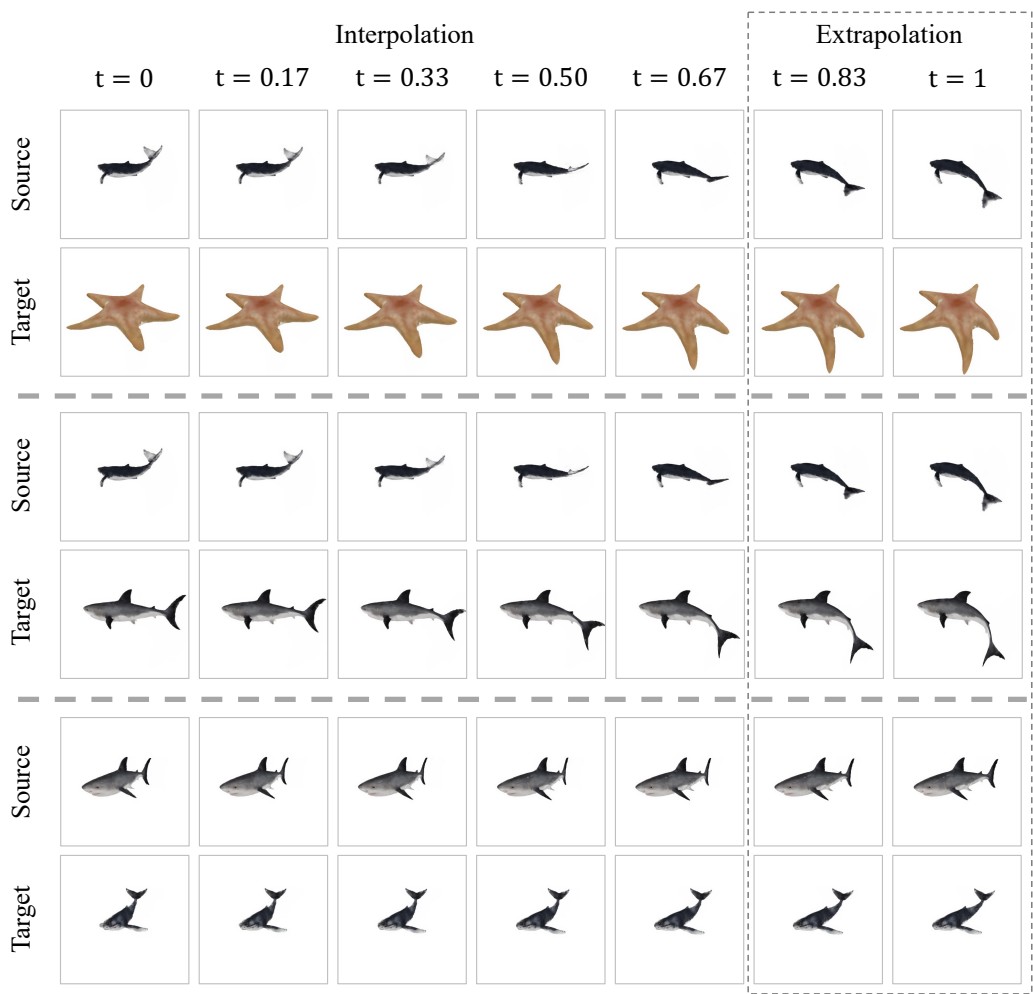

Figure 13: Qualitative results for objectwise motion transfer.

