# OpenReview forum: "NVFi: Neural Velocity Fields for 3D Physics Learning from Dynamic Videos"
_NeurIPS.cc/2023/Conference — NeurIPS 2023 poster_

### Official Review · Reviewer_3vYE · 2023-06-20

**Soundness:** 2 fair
**Presentation:** 3 good
**Contribution:** 2 fair
**Rating:** 4
**Confidence:** 5

**Summary:**

This paper presents an algorithm for the realization of novel view synthesis in dynamic scenarios, leveraging multi-view video data. In order to tackle this task, the paper introduces two key components: a keyframe dynamic field and an interframe velocity field. These fields serve the purpose of accurately representing the motion, geometry, and color information inherent in the recorded scenario. By incorporating these fields into the algorithm, the proposed approach aims to achieve effective synthesis of new views in dynamic scenarios.






**Strengths:**

This paper introduces two novel 3D datasets for dynamic object scenarios and dynamic indoor scenarios. Given the existing limitations in available datasets for view synthesis in dynamic scenarios, the introduction of these new datasets is expected to significantly enhance the progress and development of algorithms in this field. Additionally, while this paper presents innovative contributions, it also incorporates comprehensive experiments to support its claims. Furthermore, the paper is well-written and maintains a coherent structure, making it easily understandable and accessible to readers.






**Weaknesses:**

The primary concept explored in this paper involves the acquisition of a key-frame representation and the establishment of connections between key frames and intra-frames through the utilization of intraframe velocity. However, it is crucial to acknowledge that the notion of learning a canonical field and a deformation field has been previously introduced and extensively discussed in several notable publications. Notably, papers such as "D-NeRF: Neural Radiance Fields for Dynamic Scenes" (with over 500 citations) and "Nerfies: Deformable Neural Radiance Fields" (also with over 500 citations) have extensively addressed and explored this concept. The concept of learning a deformation field has also been extensively studied and advanced in previous research, as evidenced by the paper "Dynamic View Synthesis from Dynamic Monocular Video" which has accumulated over 100 citations. Moreover, while this paper introduces two novel datasets, it is important to note that there exist minimal distinctions between the proposed datasets and preexisting ones such as the Nvidia dataset, Nerfie dataset, and Iphone dataset.










**Questions:**

### Performance-related questions:

1. When compared to previous algorithms, such as D-Nerf, which rely on a canonical field, this paper employs Hexplane as a backbone. Is the improvement observed in this algorithm attributed to the utilization of Hexplane?
2. In comparison to HexPlane, which employs a single representation to learn features for the entire scenario, this paper learns multiple key-frame representations. Does the enhancement in performance stem from the usage of multiple representations instead of just one?
### Innovation-related questions:

1. Instead of directly learning the interframe velocity field, the proposed algorithm initially learns an acceleration field. Why was this approach chosen, and what evidence supports this decision?
2. Instead of learning the precise positions of key-frames, this paper employs uniform sampling for key-frame selection. Does this decision align with the inherent characteristics of dynamic scenarios, considering that motion within such scenarios may not exhibit uniform speeds?

### Dataset-related question:

1. What are the advantages of the proposed datasets in comparison to previous datasets used in similar studies?


**Limitations:**

The primary limitation of this paper revolves around the level of innovation exhibited by the proposed algorithm and the dissimilarity between the proposed datasets and their predecessors.

---

> ### Author Rebuttal · Authors · 2023-08-09
>
> **Q1:  ... D-NeRF ... Nerfies ... explored this concept.**
>
> **A:** We agree with the reviewer that both prior works are pioneering in this field. Nevertheless, we observe that only learning the deformation actually does not truly understand the physical motions. That means they are good at interpolation but fail at extrapolation. Therefore, we focus on learning a velocity field.
>
> Compared to the deformation field, the velocity field has several advantages. The first is cycle consistency. A deformation field is always unidirectional - backwardly or forwardly. If we want both deformation fields to be cycle consistent, we need to add another loss to regularize. However, we can simply integrate forwardly or backwardly on one single velocity field to get a consistent position. Secondly, the motion modeled by the velocity field can be accumulated and is continuous. This means we can easily model the difference between any two timestamps.
>
> With these motivations, we focus on designing an effective framework to estimate the underlying velocity field instead of the deformation field. In order to show the effectiveness, we introduce D-NeRF and TiNeuVox (another SOTA canonical space and deformation model, which outperforms Nerfie in many cases) as our baselines. Extensive experiments demonstrate the superiority of our learned velocity fields over deformation fields. We will include these explanations in the next version.
>
> **Q2: ... "Dynamic View Synthesis from Dynamic Monocular Video" ...**
>
> **A:** Thanks for suggesting this paper. It is a successful method that combines a dense-frame model and scene flows. However, the used scene flow can only learn motions for one step/frame. It is not a continuous motion, and there is no way to regularize future (unseen) flows. The reason we do not include this method as our baseline is that it requires depth as input. It relies on too strong priors and is not general enough.
>
> **Q3: Moreover ... Iphone dataset.**
>
> **Q4: What are the advantages ... in similar studies?**
>
> **A:**
> The primary goal of our method is to model dynamic 3D scenes for future frame extrapolation, by means of learning underlying physical velocities (*i.e.*, meaningful motions). Because most dynamic scenes in existing datasets are usually chaotic and lack predictable physical movements. We turn to propose two synthetic datasets with diverse and predictable motion patterns in the main paper.  Thanks for suggesting the new real-world NVIDIA Dynamic Scene dataset [1]. The Table 1 in Author Rebuttal shows the superiority of our method for future frame extrapolation.
>
> **Q5: When compared to ... utilization of Hexplane?**
>
> **A:** This is a great point and our extensive experiments can respond to this question in the following two aspects. Firstly, we include HexPlane based model (HexPlane\_PINN) as our baseline, whose performance is worse than ours, so our effectiveness is not just because of a new backbone. Secondly, for those scenes successfully reconstructed by D-NeRF, we can notice a huge gap between interpolation and extrapolation in Appendix Table 6. As shown in Appendix Figures 6-10, D-NeRF also predicts wrong motions in the extrapolation task. Based on this, the main reason why D-NeRF fails is due to its misunderstanding in motions
>
> **Q6: In comparison to HexPlane ... instead of just one?**
>
> **A:**
> A HexPlane model includes feature tensors in dimensions XY, XZ, YZ, XT, YT, ZT, and all of these feature tensors form the single representation. In our method, we take the same representation as HexPlane. The main difference is that the T dimension in HexPlane is always half of the total frames, and for interframes, they use bilinear interpolation to get the values, while the T dimension in VeRF is the number of keyframes, which is much fewer than HexPlane, and we use velocity field to transport interframes to keyframes. In general, both our model and HexPlane employ a single representation.
>
> **Q7:  ... an acceleration field ... decision?**
>
> **A:** For clarity, we do not initially learn an acceleration field. We learn it only by our physics losses along with learning the velocity field simultaneously. Here we briefly discuss why this is needed. Usually, the hidden velocity is not a constant velocity. For example, in a falling ball scene, there is gravitational acceleration, and in scenes with rotation, there is centripetal acceleration. In order to learn a reasonable velocity field following physics rules, we must estimate an accompanied acceleration field to guide our extrapolation. Note that, we do not add any priors and supervision to these accelerations. The only gradients passing through these accelerations are from the momentum conservation PINN loss. More details about the PINN loss are in Appendix A.4.
>
> We will clarify this point in the next version.
>
> **Q8: Instead of ... uniform speeds?**
>
> **A:** There are two reasons why we use uniform distributed keyframes. Firstly, uniform keyframes can also handle non-uniform motions. For example, in our datasets, the falling ball scene has a gravitational acceleration, and for scenes such as telescope, their rotations require a centripetal acceleration. None of them is uniform motion, and either the direction or the magnitude of the velocity keeps changing over time. As shown in the experiments, our model works well in these cases.
>
> Secondly, uniform keyframes have consistent time intervals between two keyframes. Thus we can easily control the integration step (influence GPU memory and model performance) and the integration stability (influences model performance). However, if the time intervals between keyframes are different, in order to control the GPU memory for a longer integration, we need to sacrifice the integration stability. We evaluate our model with random keyframe places on our Dynamic Scene dataset. Not surprisingly, the Table 6 in Author Rebuttal shows a decrease in performance. We will add these results in the next version.

---

> > ### Comment · Reviewer_3vYE · 2023-08-18
> > **Reply to rebuttal**
> >
> > Thank you very much for your detailed responses and valuable contributions, which undoubtedly contribute to the advancement of our community's understanding. I am truly grateful for your clarification regarding the representation employed in your research. However, I do have a couple of concerns that I would like to address.
> >
> > Firstly, I was wondering if your method has been evaluated on a more extensive and realistic dataset, such as the ones utilized in Nerfies and the original Hexplane paper, like the Plenoptic Video dataset. As the Nvidia dataset, you proposed appears to comprise only a limited number of frames for each scenario, I believe that testing on a broader range of real-world data could provide more comprehensive insights.
> >
> > Secondly, I would like to kindly suggest that you consider reviewing the paper titled "Temporal-MPI: Enabling Multi-Plane Images for Dynamic Scene Modelling via Temporal Basis Learning," which also explores the concept of interpolation. It might be interesting to examine and potentially compare your approach with the ideas presented in that work.
> >
> > Once again, thank you for your time and efforts, and I look forward to any further insights you might provide on these matters.

---

> > > ### Author Response · Authors · 2023-08-19
> > >
> > > **Q1: Firstly, I was wondering if your method has been evaluated on a more extensive and realistic dataset, such as the ones utilized in Nerfies and the original Hexplane paper, like the Plenoptic Video dataset. As the Nvidia dataset, you proposed appears to comprise only a limited number of frames for each scenario, I believe that testing on a broader range of real-world data could provide more comprehensive insights.**
> > >
> > > **A:** Thank you for suggesting Nerfies dataset and Hexplane dataset (i.e., DyNeRF dataset).
> > >
> > > The Nerfies dataset mainly focuses on human face reconstruction from selfies. The primary challenge in this dataset is to address the slight movements between selfies, where the corresponding pixels in different images do not intersect at the same spatial locations. Since the dataset lacks a `time' dimension, it does not provide the necessary temporal information for our model to estimate a velocity field and make extrapolations.
> > >
> > > For the Hexplane Plenoptic Video dataset, we have already evaluated our method on it, which we refer to as the DyNeRF dataset in our previous rebuttal. Please see Figure 3 in the attached PDF for our evaluation results. Unfortunately, this dataset appears not suitable for our model to perform future extrapolation due to the presence of scenes involving random-like motions, such as a man pouring coffee or flaming a steak. In these scenes, the hand and the coffeemaker/flame exhibit chaotic dynamics, making it nearly impossible to learn meaningful physical velocities. As a result, our method merely predicts static frames for future extrapolation.
> > >
> > > With regard to your inquiry about why a subset of frames from the NVIDIA dataset is evaluated, we would like to clarify that we aim to present a fair comparison of our model's extrapolation capability. The omitted frames in the NVIDIA dataset largely contain random-like and unpredictable motions, which are not suitable for our evaluation. Examples of these random-like motions include the ``yeah pose" and the randomly-waving hands of the skater, as well as the sudden disappearance of other cars and the appearance of humans in the truck scenes. Since these motions are hardly predictable/extrapolated and fall outside the scope of this paper, they are naturally excluded for fair and meaningful comparisons.
> > >
> > > Overall, we appreciate the reviewer's interest in dynamic videos with random-like motions. Nevertheless, the primary goal of our method is to model 3D scenes with physically meaningful dynamics and motions which are particularly common in many robotic applications, such as catching a flying ball and avoiding a moving obstacle.
> > >
> > > **Q2: Secondly, I would like to kindly suggest that you consider reviewing the paper titled ``Temporal-MPI: Enabling Multi-Plane Images for Dynamic Scene Modelling via Temporal Basis Learning," which also explores the concept of interpolation. It might be interesting to examine and potentially compare your approach with the ideas presented in that work.**
> > >
> > > **A:** Thank you for recommending the related paper Temporal-MPI. Upon carefully examining it, we find that it introduces a method for learning temporal codes for all observed timestamps. Similar to other baselines, it falls short in its ability to extrapolate to unseen time stamps, as it lacks a mechanism to predict novel temporal codes. In this regard, it is not suitable for the extrapolation task we tackle in this paper. To respect the prior art, we will include and discuss Temporal-MPI in the related work section in the next version, specifically under the subheading of dynamic 3D representations.

---

> > > > ### Comment · Reviewer_3vYE · 2023-08-20
> > > > **Response to Comments**
> > > >
> > > > Dear Authors,
> > > >
> > > > I extend my gratitude for your comprehensive response and the intriguing concept you've presented. While I acknowledge the effort you've invested and the idea you proposed, I must note that, in light of the limitations you've outlined, I maintain my initial assessment. The algorithm's susceptibility to intricate situations such as uniform motion between frames, coupled with the fact that the datasets primarily consist of simulated scenarios, reinforces my original evaluation.
> > > >
> > > > I want to clarify that my intention isn't to strongly reject your paper. I appreciate the effort and the demonstration you've provided. My suggestion is that there's room for improvement to make the paper applicable to a wider range of situations. Overall, I find some interesting concepts in your paper, even it still needs a bit more work to meet the acceptance criteria.

---

> > > > > ### Author Response · Authors · 2023-08-21
> > > > >
> > > > > Thank you for your continued engagement with our work and the valuable feedback. We would like to address the concerns you raised regarding uniform motion and simulated datasets in our evaluation.
> > > > >
> > > > > - **Motion in our dataset:** As we previously discussed, the motion in our dataset is not uniform, but captures complex dynamics with underlying accelerations, such as gravity and centripetal acceleration. These accelerations introduce non-uniformities in the motion between frames.
> > > > >
> > > > > - **Synthetic datasets:** For the synthetic datasets which we use in the initial submission, the primary goal is to validate our model's ability to comprehend meaningful physics-based motions, thanks to the availability of precise and controllable physical properties in the simulator. Such a practice is particularly common and acceptable, especially in the early exploration of an interesting problem.
> > > > >
> > > > > - **Real-world dataset:** In our rebuttal materials, we present results on the real-world NVIDIA dataset, demonstrating our model's ability to learn real physical motions, further reinforcing its capacity to capture and extrapolate complex dynamics. This shows very encouraging potential toward the ultimate goal of applying our approach to diverse real-world scenarios in the future.
> > > > >
> > > > > We hope this could address your last concerns. We highly appreciate your feedback and will take it into consideration in our next version.

---

### Official Review · Reviewer_8aAM · 2023-07-04

**Soundness:** 3 good
**Presentation:** 3 good
**Contribution:** 3 good
**Rating:** 7
**Confidence:** 3

**Summary:**

This paper proposes a new representation of dynamic scenes, using keyframe NeRF + Velocity Field.
This representation disentangles appearance & geometry from velocity, which allows many exciting applications like future frame extrapolation, unsupervised semantic 3D scene decomposition, and dynamic motion transfer.
Obtaining such representation from videos requires a carefully designed system and physics-constrained losses.
To validate this model, this paper collects two new synthetic datasets and show impressive results on these datasets.

**Strengths:**

1. Learning a velocity field from videos is an exciting direction and could be impactful for future research, especially for physics-aware 3D/4D tasks. This paper makes an impressive attempt in this direction.
2. The method is well-motivated and intuitively reasonable. The whole idea is easy to follow and sound.
3. The results are impressive, and the applications of the velocity fields are exciting, especially the 3D semantic field segmentation results could clearly show different parts of objects.

**Weaknesses:**

1. Lacking limitation discussions. The current paper clearly has several limitations and they are expected to be discussed in detail in the paper. (1) As briefly mentioned in the broad impact, this paper doesn't have real-world data to validate whether it could work on real scenes. (2) This method seems to require videos with many views at the same timestamp as inputs. I am curious how this method works with monocular videos/sparse views. (3) How does the proposed method deal with abrupt motions and non-typology changes? For summary, the paper should discuss the potential limitations in detail.
2. Some presentations could be improved. It would be nice to give a brief explanation about losses and PINN used in this paper. And Algorithm 1 table could be more precise. The current paper seems to omit too many technical details and the meaning of some losses is confusing.

**Questions:**

I am generally positive about this paper but I think the limitations of this paper (and the questions in that bullet) should be discussed in detail.

**Limitations:**

The current version doesn't give a convincing discussion about limation and potential broad impact. See the weakness for detailed comments.

---

> ### Author Rebuttal · Authors · 2023-08-09
>
> **Q1: Lacking limitation discussions. The current paper clearly has several limitations and they are expected to be discussed in detail in the paper. (1) As briefly mentioned in the broad impact, this paper doesn't have real-world data to validate whether it could work on real scenes.**
>
> **A:** As also suggested by other reviewers, we conduct additional experiments on the real-world NVIDIA Dynamic Scene dataset. It captures real-world dynamic scenes by a static camera rig with 12 cameras. For each scene, we clip 60 frames with reasonable and predictable motions. We reserve the first 46 frames at randomly picked 11 cameras as the training split, *i.e.*, 506 frames, while leaving the 46 frames at the remaining 1 camera for testing interpolation ability, *i.e.*, 46 frames for novel view synthesis within the training time period, and keeping the last 14 frames at all 12 cameras for evaluating future frame extrapolation, *i.e.*, 168 frames. As shown in the Table 1 in Author Rebuttal, our method achieves significantly better results in the challenging task of future frame extrapolation. Figure 1 shows the qualitative results in the appended PDF. Due to the time limit, we can only provide scores on two scenes. More experiments are still running and we will add complete results in the next version.
>
>
> **Q2: (2) This method seems to require videos with many views at the same timestamp as inputs. I am curious how this method works with monocular videos/sparse views.**
>
> **A:** As requested, we additionally evaluate our method on the truck scene from NVIDIA Dynamic Scenes in a monocular way. In particular, for every timestamp, only one camera from the 12 cameras is used. Due to the depth ambiguity of the monocular video, it is very hard to disentangle the foreground and background of the scene. So the velocity field will influence the background scene as illustrated in Figure 2 in the appended PDF. We leave this challenging monocular setting for our future exploration.
>
> As suggested by reviewer c2wX, we also make an ablation study on the camera number on our own Dynamic Objects dataset. The Table 3 in Author Rebuttal shows the quantitative results. As expected, given fewer training camera views, the performance of our method drops sharply, primarily because the extremely sparse views are unlikely to capture sufficient visual information for physical motion learning.
>
>
> **Q3: (3) How does the proposed method deal with abrupt motions and non-typology changes?**
> **A:** This is a good question. We evaluate our model on two scenes from DyNeRF dataset. Since the abrupt motions of arm and flame / coffeemaker in hands are actually not predictable, we clearly fail on the extrapolation task, even though we can get promising results in interpolation. Figure 3 of the appended PDF shows the qualitative results.
>
> Above all, our model aims to learn the predictable physical dynamics, and is not able to predict abrupt motions. We will discuss this limitation in the next version.
>
>
> **Q4: Some presentations could be improved. It would be nice to give a brief explanation about losses and PINN used in this paper**
>
> **A:** We thank the reviewer for this suggestion. To explain the losses better, we first need to address that the whole dynamics is regarded as a transport problem, as shown in Appendix A.4. In particular, the density and appearance features of objects are transported by a velocity field, and the velocity field is transported by itself according to some unobserved hidden forces. So the losses can be divided into two types. The first type is the RGB rendering loss for both keyframes and interframes, which are MSE between the rendering pixel colors and the ground-truth pixel colors.
>
> The second type is the PINN PDE losses, where the divergence-free loss is used to constrain the mass conservation of objects in the scene, and the momentum conservation loss is to learn the hidden forces and ensure the extrapolation in a reasonable pattern.
>
> PINN losses are implemented as follows: 1) We uniformly sample points in time and space dimensions. 2) We use torch.autograd to evaluate the jacobian of velocity w.r.t the input position and time. 3) We put those terms to calculate the LHS of Equation 7 and make it an L2 loss. Moreover, since the acceleration (hidden forces) is not observable, it is totally estimated by PINN losses.
>
> We will add these explanations in the next version.
>
> [1] J. S. Yoon. et al, Novel View Synthesis of Dynamic Scenes with Globally Coherent Depths from a Monocular Camera. CVPR, 2020.

---

> > ### Comment · Reviewer_8aAM · 2023-08-16
> > **Reply to Rebuttal**
> >
> > Thanks for the explanation. After reading all reviews and rebuttals, I think the rebuttal address my concerns and I improve my final rating to acceptance.

---

> > > ### Author Response · Authors · 2023-08-18
> > >
> > > We highly appreciate the reviewer's time in reviewing our rebuttal materials and providing very positive feedback. We also thank your initial comments which clearly improve our manuscript.

---

### Official Review · Reviewer_hQnr · 2023-07-06

**Soundness:** 3 good
**Presentation:** 1 poor
**Contribution:** 3 good
**Rating:** 6
**Confidence:** 5

**Summary:**

This paper proposes a model for dynamic 3D scenes from multi-view videos. Different with previous method, this paper proposes a physical velocity field instead of a deformation field. The method uses a dynamic radiance field to model key frames, and use a velocity field to warp the key frame to intermedia frames. For optimization, the photometric loss of rendered keyframes and interframes are optimized together, along with several PINN terms. Also, the authors show some applications of their model: future frame extrapolation, semantic 3D scene decomposition, dynamic motion transfer. Two new synthetic datasets are introduced.

**Strengths:**

1.	Although velocity field has been introduced to this area (Neural Radiance Flow for 4D View Synthesis and Video Processing), this paper introduces a more complete solution, including the PINN terms and the warping mechanism.
2.	The implementations of T-NeRF_PINN and HexPlane_PINN are not clear to me, but the future frame extrapolation of the proposed method is impressive.
3.	Two new synthetic datasets are proposed.

**Weaknesses:**

1.	The method shows inconsistent performance on interpolation.
2.	For dense frame optimization method, T-NeRF is not an appropriate method to compare with, since it is a very naïve solution which simply adds a time dimension to MLP input. To prove the effectiveness of the proposed method, I think NSFF (Neural Scene Flow Field) is a reasonable and meaningful method to compare. NSFF also has warping constraints.
3.	The proposed method is basically a canonical based method which separate the whole video into multiple sections and assign a canonical model for each section. From this aspect, it is not so fair to compare with TiNeuVox in this way. How would TiNeuVox perform if also assign the same number of canonical models to it?
4.	The potential application of motion transfer is not clear to me.
5.	No interpolation performance for ablation study.
6.	No experiment on real dataset. NHR dataset (Multi-view Neural Human Rendering) may be an appropriate dataset for this paper.
7.	From my aspect, the authors spent too much space on applications. The performance of the interpolation is more important for me to identify the ability of the proposed method to model the dynamic scenes.

**Questions:**

1.	For motion transfer, how would the learnt velocity field align with new objects? Put original images of Gnome in Figure 3 (c) may help understanding.

**Limitations:**

The authores addresed the limitations, including no experiment on real dataset.

---

> ### Author Rebuttal · Authors · 2023-08-09
>
> **Q1: The method shows inconsistent performance on interpolation.**
>
> **A:** Comprehensive and consistent results for the interpolation of our method are provided in the Appendix. We are not clear about the request of this comment, but we are always open to discussion with the reviewer.
>
>
> **Q2: ... T-NeRF is not an appropriate method to compare with, ... I think NSFF (Neural Scene Flow Field) is a reasonable and meaningful method to compare.**
>
> **A:** Thanks for the valuable suggestion. We add NSFF as a new baseline on our Dynamic Indoor Scene dataset, as NSFF is not suitable for white backgrounds in our Dynamic Object dataset. As shown in the Table 4 in Author Rebuttal, NSFF indeed obtains much better interpolation results than the naive T-NeRF, but it still fails to extrapolate future frames. Figure 4 shows the qualitative results in the appended PDF. We will add this new baseline in the main paper in the next version.
>
>
> **Q3: ... How would TiNeuVox perform if also assign the same number of canonical models to it?**
>
> **A:** This is a very good point to discuss. TiNeuVox is a deformation field based model. If we use several canonical spaces, each canonical space requires a distinct deformation field to deform the latter frames back to it. This means that using multiple canonical spaces is equivalent to slicing the dataset into several pieces, and several TiNeuVoxs are trained on each piece respectively. Then the extrapolation is only related to the final canonical space and deformation field. This strategy actually cannot significantly increase the ability of TiNeuVox.
>
> To verify this, we conduct additional experiments for TiNeuVox with multiple canonical spaces. From the Table 5 in Author Rebuttal, we can see that TiNeuVox with the same 16 canonical spaces still fails to obtain satisfactory extrapolation results as ours, showing the superiority of our learning of physical velocity fields. Due to time limit, more experiments are still running and we will include complete results in the next version.
>
>
> **Q4: The potential application of motion transfer is not clear to me.**
>
> **A:** One interesting application is character animation using motion transfer. There are some demos in Appendix Figure 12 and in the final part of the demo. Another potential application is to edit the reconstructed scene. For example, we can replace the objects with other ones, which are not in the original dynamic scene. Unarguably, there could be many other exciting use cases and we hope that our method could unlock new opportunities.
>
>
> **Q5: No interpolation performance for ablation study.**
>
> **A:** The interpolation performance is in Appendix Table 5. Thanks for your reminder and we will include this in the main paper in the next version.
>
>
> **Q6: No experiment on real dataset. NHR dataset (Multi-view Neural Human Rendering) may be an appropriate dataset for this paper.**
>
> **A:** Thanks for the suggestion. We find that NHR dataset is a point cloud rendering dataset, which is not suitable for our method. Alternatively, we evaluate our method on the real-world NVIDIA Dynamic Scene dataset[1]. The Table 1 in Author Rebuttal shows the superiority of our method for future frame extrapolation.
>
>
> **Q7: From my aspect, the authors spent too much space on applications. The performance of the interpolation is more important for me to identify the ability of the proposed method to model the dynamic scenes.**
>
> **A:** We agree that interpolation is indeed an important task, which can be seen from a large number of existing research works in recent two years.  In the meantime, we also strongly argue that the future frame extrapolation ability is essential for many intelligent machines, and related research is still in its infancy. Unfortunately, existing interpolation techniques fail to predict future frames as shown in our experiments. In this regard, we hope that our proposed method could inspire more advanced works in the future.
>
>
> **Q8 : For motion transfer, how would the learnt velocity field align with new objects? Put original images of Gnome in Figure 3 (c) may help understanding.**
>
> **A:** A naive motion transfer requires objects to have similar sizes. This is illustrated by Figure 6 in the attached PDF and the video demo Motion Transfer section. For a general motion transfer, more advanced techniques such as shape registration and alignment may be applied to deal with variable sizes of objects.
>
>
> **Q9: The implementations of T-NeRF$ _{PINN} $ and HexPlane$ _{PINN} $ are not clear to me, but the future frame extrapolation of the proposed method is impressive.**
>
> **A:** T-NeRF_pinn is implemented as an original T-NeRF along with a velocity field (MLPs) which is the same as our VeRF. HexPlane_pinn is implemented as a HexPlane along with the same velocity field (MLPs). Different from VeRF, the two models are trained as PINN.
>
> First of all, the RGB loss of given supervision is used to train the density and color. In PINN framework, it can be also regarded as boundary conditions. Then we regard the whole dynamics as a transport problem, and more details are in Appendix A.4. We add the transport loss for density and colors, and the physics loss for the velocity field as PINN PDE constraints. The main difference between these two models and VeRF is that, in VeRF, gradients can flow through the velocity field from the RGB loss, while in *$_{PINN}$ models the velocity field is only estimated by PINN losses.
>
> We will add these explanations in the next version.
>
> [1] J. S. Yoon. et al, Novel View Synthesis of Dynamic Scenes with Globally Coherent Depths from a Monocular Camera. CVPR, 2020.

---

> > ### Comment · Reviewer_hQnr · 2023-08-14
> > **Impressive results for extrapolation, but less competitive performance for interpolation**
> >
> > Thanks for the hard working to address my concerns. Now, the only concern that remains to prevent me from recommending the paper to be accepted is the relative incompetitive performance for interpolation. As the paper title suggested, this is a method to representing dynamic 3D scenes. I intend to see at least competitive performance to model the dynamic 3d scenes. Also, I notice that the LPIPS of the extrapolation is not always the best in Table 1 and Table 5, which does not match the huge difference of the PSNR. What may be the reason?

---

> > > ### Author Response · Authors · 2023-08-18
> > >
> > > **Q: Thanks for the hard working to address my concerns. Now, the only concern that remains to prevent me from recommending the paper to be accepted is the relative incompetitive performance for interpolation. As the paper title suggested, this is a method to representing dynamic 3D scenes. I intend to see at least competitive performance to model the dynamic 3d scenes.**
> > >
> > > **A:** We agree with the reviewer that interpolation is also important in modeling dynamic 3D scenes. As requested, we turn to retraining our models using new settings from scratch on the Dynamic Indoor Scene dataset and NVIDIA Dynamic Scene dataset.
> > >
> > > In particular, on our Dynamic Indoor Scene dataset, we just use 4 keyframes instead of 16, while keeping all other settings untouched. As shown in the following Table 7, our method with the new number of keyframes outperforms the strong baseline TiNeuVox in both interpolation and extrapolation. Basically, using fewer keyframes results in more interframes to supervise each keyframe, enhancing scene reconstruction in cluttered environments; for instance, with 4 keyframes instead of 16, each keyframe is supervised by 4 times as many images, providing more comprehensive coverage of occluded areas and better interpolation from novel views (interframes). Additionally, fewer keyframes require longer integration of motion, which, in indoor scenes characterized by rigid body motions, may result in more stable motion and consequently improved geometry reconstruction. The ablation study in Author Rebuttal Table 2 also shows a similar trend. Nevertheless, searching for an optimal setting of keyframes needs more experiments.
> > >
> > > On the real-world NVIDIA Dynamic Scene dataset which is much more challenging, we turn to using 12M grids instead of 8M grids for the spatial resolution of the backbone Hexplane, 120K instead of 60K iterations for longer training, and 1024 instead of 2048 sampling rays for better Cuda memory management. As shown in the following Table 8, our method with the new settings achieves comparable performance with TiNeuVox in interpolation, along with significantly better results in extrapolation. Due to the time limit, only the Skating scene has been evaluated.
> > >
> > > Above all, we primarily focus on extrapolation and have not well-searched training settings for the task of interpolation in our main paper. From the new experiments above, we can see that our method actually does not scarify the accuracy of interpolation. Instead, it can achieve superior performance in both interpolation and extrapolation given better choices of training hyper-parameters.
> > >
> > > In addition to adding the above new experimental results in our next version, we also consider changing the paper title to ``Learning Velocity Fields for Dynamic 3D Scenes from Multiview Videos" if the reviewer agrees to it.
> > >
> > > We look forward to the reviewer's new feedback.
> > >
> > > **Table 7: Quantitative results of our method and TiNeuVox on Dynamic Indoor Scene dataset. Ours$^\*$ represents the new setting $K=4$.**
> > > |                      |            | Interpolation |           |            | Extrapolation |           |
> > > |----------------------|------------|---------------|-----------|------------|---------------|-----------|
> > > | **Models**               | **PSNR**       | **SSIM**          | **LPIPS**     | **PSNR**       | **SSIM**          | **LPIPS**     |
> > > | TiNeuVox             | 29.981     | 0.864         | 0.213     | 21.029     | 0.770         | 0.281     |
> > > | VeRF (Ours, K=16)    | 28.000     | 0.862         | 0.226     | 26.235     | 0.839         | 0.237     |
> > > | VeRF (Ours, K=4)$^\*$ | **30.675** | **0.877**     | **0.211** | **29.745** | **0.876**     | **0.204** |
> > >
> > > **Table 8: Quantitative results of our method and TiNeuVox on Skating scene from the NVIDIA Dynamic Scene dataset. Ours$^\*$ represents the new settings.**
> > > |                             |            | Interpolation |           |            | Extrapolation |           |
> > > |-----------------------------|------------|---------------|-----------|------------|---------------|-----------|
> > > | **Models**               | **PSNR**       | **SSIM**          | **LPIPS**     | **PSNR**       | **SSIM**          | **LPIPS**     |
> > > | TiNeuVox                    | **29.377** | **0.889**     | 0.202     | 24.224     | 0.878         | 0.220     |
> > > | VeRF (Ours, 8M grids)       | 26.999     | 0.848         | 0.227     | 28.654     | 0.896         | 0.208     |
> > > | VeRF (Ours, 12M grids)$^\*$ | $\underline{29.064}$     | $\underline{0.888}$         | **0.193** | **29.026** | **0.898**     | **0.193** |

---

> > > > ### Comment · Reviewer_hQnr · 2023-08-22
> > > >
> > > > Thanks for your further experiments. I would like to increase my rating.

---

> > > ### Author Response · Authors · 2023-08-18
> > >
> > > **Q: Also, I notice that the LPIPS of the extrapolation is not always the best in Table 1 and Table 5, which does not match the huge difference of the PSNR. What may be the reason?**
> > >
> > > **A:** Since these numbers compare the extrapolation performance instead of interpolation, multiple factors may cause this issue. For example, in training, there are some areas that are not observable, which leads to some blank areas after extrapolation. The impact of this blank area will be increased in cognitive models' feature space. That's potentially why the LPIPS could be smaller but PSNR higher.

---

### Official Review · Reviewer_c2wX · 2023-07-07

**Soundness:** 3 good
**Presentation:** 4 excellent
**Contribution:** 3 good
**Rating:** 7
**Confidence:** 3

**Summary:**

The paper aims to model dynamic 3D scenes from multiple-videos. The method simultaneously learns for geometry, appearance, and physical velocity of the 3D scenes from video frames. The show different application like future frame extraploation, unsupervised semantic 3D scene decomposition and dynamic motion transfer using the current framework. They further proposed two dynamic 3D datasets with extensive experiments.

**Strengths:**

The paper uses three major components to tackle the dynamic scene,
the first is using the keyframe dynamic radiance field to compute radiance field at different time instances would be a good initial 3D estimator.

the second is to use velocity field to interpolate the intermediate frames instead of recomputing NERF per frame which is very expensive.

Finally a joint keyframe and velocity based interframe optimization produces an smooth trajectory and reconstruction of the dynamic scene.

Instead of just showcasing the advantage of the method, displaying the robustness of the algorithm to multiple downstream tasks like future frame Extrapolation signifies the advantage of this method.

Good ablation study and the introduced datasets are helpful for future research in dynamic scene understanding.

**Weaknesses:**

There has been many datasets proposed to tackle the problem of dynamic scene understanding, What is the advantage or difference with respect to these datasets has not been well studies. Specifically, Line 187 states that all of the datasets currently available are not useful for this task. It would be interesting to see the effect of the proposed method on one these datasets to understand the advantages and the disadvantages of the proposed algorithm.

Effect of choosing keyframe is specific to the dataset. Since the current dataset has only one moving object may be using 16 frames is sufficient. The ablation should show more experiments to justify the keyframe selection process.

The effect of the number of cameras is not explored in the analysis. Since you are using dynamic scenes, the effect of fewer frames should be analyzed.

**Questions:**

How does the method perform on other dyanmic scene datasets. even using simple multi-view datasets like panoptic studio should be explored to see the effect of the algorithm.

Are the camera poses already provided for the dataset?

**Limitations:**

limitations have been discussed

---

> ### Author Rebuttal · Authors · 2023-08-09
>
> **Q1: There has been many datasets proposed to tackle the problem of dynamic scene understanding, What is the advantage or difference with respect to these datasets has not been well studies.**
>
> **A:** The primary goal of our method is to model dynamic 3D scenes for future frame extrapolation, by means of learning underlying physical velocities (*i.e.*, meaningful motions) in the continuous 3D space. In contrast, existing dynamic 3D scene modeling techniques and the commonly used datasets in the literature are mainly designed for novel view rendering/interpolation within the training time period, rather than for extrapolation beyond the training time period. For example, the dynamic motion patterns in existing datasets such as DyNeRF dataset [2] are usually chaotic and lack predictable physical movements. This research gap motivates us to propose two synthetic datasets with diverse and predictable motion patterns in the main paper.
>
> Nevertheless, as also suggested by other reviewers, we further pick up a number of meaningful 3D scenes from the real-world NVIDIA Dynamic Scene dataset [1]. The Table 1 in Author Rebuttal shows the superiority of our method for future frame extrapolation.
>
>
> **Q2: Specifically, Line 187 states that all of the datasets currently available are not useful for this task. It would be interesting to see the effect of the proposed method on one these datasets to understand the advantages and the disadvantages of the proposed algorithm.**
>
> **A:** Thanks for the suggestion. As requested, we train our model on two typical scenes of DyNeRF dataset [2]. In these scenes, a man is pouring coffee / flaming a steak, and the hand and coffeemaker / flame are just undergoing random motions. As shown in Figure 3 in the appended PDF, not surprisingly, our method is not able to learn meaningful physical velocities and it is impossible to have correct extrapolations on these chaotic dynamics. Actually, our method simply predicts static frames for future extrapolation.
>
>
> **Q3: Effect of choosing keyframe is specific to the dataset. Since the current dataset has only one moving object may be using 16 frames is sufficient. The ablation should show more experiments to justify the keyframe selection process.**
>
> **A:** We highly appreciate this suggestion and conduct an ablation study on the number of keyframes on our Dynamic Indoor Scene dataset. As shown in the Table 2 in Author Rebuttal, we find that fewer keyframes tend to have better performance, demonstrating that our keyframe based optimization strategy is actually very flexible and effective. More experiments are still running and we will add these results in the next version.
>
>
> **Q4: The effect of the number of cameras is not explored in the analysis. Since you are using dynamic scenes, the effect of fewer frames should be analyzed.**
>
> **A:** Thank you for this advice and we conduct an additional ablation study on the number of training cameras on our Dynamic Objects dataset. The Table 3 in Author Rebuttal shows the quantitative results. As expected, given fewer training camera views, the performance of our method drops sharply, primarily because the extremely sparse views are unlikely to capture sufficient visual information for physical motion learning.
>
>
> **Q5: How does the method perform on other dynamic scene datasets. even using simple multi-view datasets like panoptic studio should be explored to see the effect of the algorithm.**
>
> **A:** As also suggested by other reviewers, we turn to evaluate our method on another real-world dataset: NVIDIA Dynamic Scene dataset [1]. Results are supplied in above Q1.
>
>
> **Q6: Are the camera poses already provided for the dataset?**
>
> **A:** Yes, all camera poses are given in the datasets. Nevertheless, it is very interesting to explore  simultaneous camera pose estimation and scene modeling in the future.
>
> [1] J. S. Yoon. et al, Novel View Synthesis of Dynamic Scenes with Globally Coherent Depths from a Monocular Camera. CVPR, 2020.
> [2] T. Li. et al, Neural 3d video synthesis from multiview video. CVPR, 2022.

---

### Official Review · Reviewer_e7NB · 2023-07-11

**Soundness:** 3 good
**Presentation:** 3 good
**Contribution:** 3 good
**Rating:** 6
**Confidence:** 4

**Summary:**

The paper presents the framework to simultaneously learn the geometry, appearance, and velocity from only video frames. The paper's contribution lies in three directions: 1) Keyframe dynamic radiance fields to learn the time-dependent volume density and appearance. 2) Interframe velocity field to learn the time-dependent 3D velocity and 3) joint keyframe and interframe optimization to train keyframe and interframe fields together with physics-informed constraints. The main contribution lies in the joint keyframe and inter-frame optimization, which introduces three loss functions that help precisely learn to disentangle object masks, types, and materials.

**Strengths:**

S1) The framework is simple to understand yet elegant in achieving many tasks.

S2) The paper is well written, with detailed derivation of major contributions. Easy to understand.

S3)  The work presents Dynamic Object and Dynamic Indoor Scene datasets.

S4) The evaluation of the proposed approach is well presented on multiple baselines and downstream tasks (Frame extrapolation, Scene decomposition, Motion transfer). The proposed approach outperforms the existing approaches.


**Weaknesses:**

W1) All the evaluation results are presented on a synthetic dataset where the motion information is constrained ideally. It is unclear from the experiments how well the results translate to real-world scenarios.

W2) All the results presented in this work are evaluated only on the authors' dataset. It is vital for the paper to present results on publicly available datasets to establish benchmarks (at least for a few downstream tasks).


**Questions:**

Q1) From qualitative results, the dataset used in this work has limited spatial resolution. So, how would the approach scale to real-looking scenes with larger resolution?


**Limitations:**

See the weaknesses section.

---

> ### Author Rebuttal · Authors · 2023-08-09
>
> **Q1: All the evaluation results are presented on a synthetic dataset where the motion information is constrained ideally. It is unclear from the experiments how well the results translate to real-world scenarios.**
>
> **Q2: All the results presented in this work are evaluated only on the authors' dataset. It is vital for the paper to present results on publicly available datasets to establish benchmarks (at least for a few downstream tasks).**
>
> **A:** Thanks for the suggestions and we agree that establishing a benchmark on a public and real-world dataset is crucial for the field of study. We evaluate our method on the real-world dataset: NVIDIA Dynamic Scene[1]. It captures real-world dynamic scenes by a static camera rig with 12 cameras. For each scene, we clip 60 frames with reasonable and predictable motions. We reserve the first 46 frames at randomly picked 11 cameras as the training split, *i.e.*, 506 frames, while leaving the 46 frames at the remaining 1 camera for testing interpolation ability, *i.e.*, 46 frames for novel view synthesis within the training time period, and keeping the last 14 frames at all 12 cameras for evaluating future frame extrapolation, *i.e.*, 168 frames. As shown in the Table 1 in Author Rebuttal, our method achieves significantly better results in the challenging task of future frame extrapolation. Figure 1 shows the qualitative results in the appended PDF. Due to the time limit, we can only provide scores on two scenes. More experiments are still running and we will add complete results in the next version.
>
>
> **Q3: From qualitative results, the dataset used in this work has limited spatial resolution. So, how would the approach scale to real-looking scenes with larger resolution?**
>
> **A:** This is a very good point. In the newly added NVIDIA Dynamic Scene dataset [1], the spatial scale of the real-world scene ``Truck" is about 20x10x10 meters, which is clearly larger than the two synthetic datasets in the main paper. From our new results, we can see that our method can easily scale up.
>
> [1] J. S. Yoon. et al, Novel View Synthesis of Dynamic Scenes with Globally Coherent Depths from a Monocular Camera. CVPR, 2020.

---

### Author Rebuttal · Authors · 2023-08-10

Firstly, we would like to thank all five reviewers for their valuable comments. We genuinely appreciate the time and effort invested in reviewing our paper. Based on the comments provided, we have made significant improvements to our paper:

- Additional experiments on public real-world datasets, including: a)
two distinct scenes derived from NVIDIA Dynamic Scene dataset, b) one scene from the monocular version of NVIDIA Dynamic Scene dataset, c) two scenes from DyNeRF dataset.

- Additional evaluation of a new baseline, Neural Scene Flow Fields (NSFF), on our Dynamic Indoor Scene dataset.

- Additional ablation studies about: a) the number of keyframes used on our Dynamic Indoor Scene dataset, b) the number of cameras used in our Dynamic Object datasets.

Due to the character limitations for responses in review rebuttal, we have cataloged all the relevant tables and details in this author rebuttal. We kindly request reviewers to refer to these tables and sections for a more detailed understanding.


**Table 1:** Quantitative results of our method and baselines on the NVIDIA Dynamic Scene dataset.
|                   |            |               |   Truck   |            |               |           |
|:-----------------:|:----------:|:-------------:|:---------:|:----------:|:-------------:|:---------:|
|                   |            | **Interpolation** |           |            | **Extrapolation** |           |
|       **Model**       |    **PSNR**    |      **SSIM**     |  **LPIPS**   |    **PSNR**    |      **SSIM**     |   **LPIPS**   |
|      TiNeuVox     |   27.230   |   **0.846**   | **0.229** |   24.887   |     0.848     | **0.209** |
| HexPlane$_{PINN}$ |   25.494   |     0.768     |   0.337   |   24.991   |     0.768     |   0.325   |
|    VeRF (Ours)    | **27.276** |     0.840     |   0.235   | **28.269** |   **0.855**   |   0.220   |
|                   |            |               |  **Skating**  |            |               |           |
|                   |            | **Interpolation** |           |            | **Extrapolation** |           |
|       **Model**       |    **PSNR**    |      **SSIM**     |   **LPIPS**   |    **PSNR**    |      **SSIM**     |   **LPIPS**   |
|      TiNeuVox     | **29.377** |   **0.889**   | **0.202** |   24.224   |     0.878     |   0.220   |
| HexPlane$_{PINN}$ |   24.447   |     0.867     |   0.225   |   23.955   |     0.868     |   0.232   |
|    VeRF (Ours)    |   26.999   |     0.848     |   0.227   | **28.654** |   **0.896**   | **0.208** |

**Table 2:** Ablation study of the keyframe number on our Dynamic Indoor Scene dataset.
|           |            | Interpolation |           |            | Extrapolation |           |
|-----------|------------|---------------|-----------|------------|---------------|-----------|
| **Keyframes** | **PSNR**       | **SSIM**          | **LPIPS**     | **PSNR**       | **SSIM**          | **LPIPS**     |
| 8         | **30.321** | **0.871**     | **0.220** | **29.093** | **0.873**     | **0.225** |
| 16        | 28.000     | 0.862         | 0.226     | 26.235     | 0.839         | 0.237     |
| 32        | 29.764     | 0.851         | 0.255     | 26.634     | 0.828         | 0.247     |

**Table 3:** Ablation study of the camera number on our Dynamic Objects dataset.
|         |            | Interpolation |           |            | Extrapolation |           |
|---------|------------|---------------|-----------|------------|---------------|-----------|
| **cameras** | **PSNR**       | **SSIM**          | **LPIPS**     | **PSNR**       | **SSIM**          | **LPIPS**     |
| 12      | **29.027** | **0.970**     | **0.039** | **27.594** | **0.972**     | **0.036** |
| 6       | 25.689     | 0.954         | 0.051     | 25.114     | 0.959         | 0.122     |
| 3       | 21.460     | 0.912         | 0.088     | 21.370     | 0.917         | 0.084     |

**Table 4:** Quantitative results of NSFF on the Dynamic Indoor Scene dataset.
|       |            | Interpolation |           |            | Extrapolation |           |
|-------|------------|---------------|-----------|------------|---------------|-----------|
| **Model** | **PSNR**       | **SSIM**          | **LPIPS**     | **PSNR**       | **SSIM**          | **LPIPS**     |
| NSFF  | 29.365     | 0.829         | 0.278     | 24.163     | 0.795         | 0.289     |
| VeRF  | **30.321** | **0.871**     | **0.220** | **29.093** | **0.873**     | **0.225** |

**Table 5:** Quantitative results of TiNeuVox on the chessboard of our Dynamic Indoor Scene dataset.
|                         |            | Extrapolation |           |
|-------------------------|------------|---------------|-----------|
| **Model**                   | **PSNR**       | **SSIM**          | **LPIPS**     |
| TiNeuVox (1 Canonical)  | 19.718     | 0.765         | 0.310     |
| TiNeuVox (16 Canonical) | 21.394     | 0.812         | **0.233** |
| VeRF (Ours, K=18)       | **24.160** | **0.837**     | 0.259     |

**Table 6:** Non-uniform sampling for key-frame selection.
|            |        | Interpolation |       |        | Extrapolation |       |
|------------|--------|---------------|-------|--------|---------------|-------|
| **Keyframes**  | **PSNR**   | **SSIM**          | **LPIPS** | **PSNR**   | **SSIM**          | **LPIPS** |
| Uniform 16 | **29.027** | **0.970**         | **0.039** | **27.594** | **0.972**         | **0.036** |
| Random 16  | 28.463 | 0.966         | 0.043 | 24.489 | 0.959         | 0.045 |

---

### Author Response · Authors · 2023-08-21
**Summary of Rebuttal Discussions**

We thank the valuable feedback from all reviewers and conduct a series of additional experiments and ablation studies to further validate and improve our method. Below is a consolidated summary of the changes made and their corresponding results:

- **New Experiments on Real-World Datasets:** We evaluate our method on the NVIDIA Dynamic Scene and DyNeRF datasets. For the NVIDIA Dynamic Scene dataset, our method achieves significantly better results in the challenging task of future frame extrapolation, as evidenced in Table 1 of the Author Rebuttal. However, on the DyNeRF dataset where motions are chaotic (e.g., a man pouring coffee/flaming a steak), our method struggles to learn meaningful physical velocities and simply predicts static frames for future extrapolation, as shown in Figure 3 of the appended PDF.

- **More Baseline Evaluation:** We include an evaluation of a new baseline, Neural Scene Flow Fields (NSFF), on our Dynamic Indoor Scene dataset. The results, as shown in Table 4 of the Author Rebuttal, highlight the advantages of our method over this state-of-the-art alternative.

- **New Ablation Study on Keyframes:** We conduct an ablation study on the number of keyframes on our Dynamic Indoor Scene dataset. Interestingly, we find that fewer keyframes tend to result in better performance (Table 2 of Author Rebuttal). Additionally, using just 4 keyframes instead of 16, our method outperforms the strong baseline TiNeuVox in both interpolation and extrapolation, as shown in Table 7 in the response to Reviewer hQnr.

- **New Ablation Study on Training Cameras:** We perform an ablation study on the number of training cameras on our Dynamic Objects dataset. The results (Table 3 in Author Rebuttal) indicate that performance drops sharply with fewer camera views, primarily due to insufficient visual information for physical motion learning.

- **More Experiments with TiNeuVox:** Additional experiments for TiNeuVox with multiple canonical spaces reveal that our method still outperforms TiNeuVox in terms of learning physical velocity fields (Table 5 of Author Rebuttal).

- **New Evaluation on Monocular Videos:** We evaluate our method on the truck scene from the NVIDIA Dynamic Scene dataset in a monocular setting. The results (Figure 2 in appended PDF) demonstrate challenges due to depth ambiguity, affecting the disentanglement of foreground and background.

- **New Evaluation on Random Keyframe Selection:** Our evaluation with random keyframe selection on the Dynamic Scene dataset shows a decrease in performance (Table 6 of Author Rebuttal).

- **Better Training Settings for both Interpolation and Extrapolation:** On the real-world NVIDIA Dynamic Scene dataset, we opt for better training settings, achieving comparable performance to TiNeuVox in interpolation and significantly better results in extrapolation (Table 8  in the response to Reviewer hQnr). Our method demonstrates superior performance in both interpolation and extrapolation with better choices of training hyperparameters.

In summary, the additional experiments and ablation studies have provided further evidence of the robustness and effectiveness of our method. We hope that our rebuttal materials can adequately address all your concerns and contribute to a more comprehensive evaluation of our work.

---

### Decision · Program_Chairs · 2023-09-21

**Decision:**

Accept (poster)

**Comment:**

The paper presents a new representation for modeling dynamic 3D scenes. The initial reviews were overall positive. There were concerns about the lack of comparisons with other state-of-the-art dynamic NeRF methods and real-world videos. The authors' rebuttal provided the comparisons with TiNeuVox and results on two datasets, effectively alleviating these initial concerns. After discussions, the reviewers converge to accept the paper.